# A Survey on Benchmarks of LLM-based GUI Agents

**Yihong Chen**                                                    *chenyihong@mail.tsinghua.edu.cn*
*Department of Electronic Engineering*
*Tsinghua University*

**Shuai Wang**                                                    *shuai.wang.sw2572@yale.edu*
*Yale University*

**Yaqing Wang**                                                    *wangyaqing@bimsa.cn*
*Beijing Institute of Mathematical Sciences and Applications*

**Quanming Yao**                                                    *qyaoaa@tsinghua.edu.cn*
*Department of Electronic Engineering, Tsinghua University*
*Beijing National Research Center for Information Science and Technology*
*State Key Laboratory of Space Network and Communications*

**Reviewed on OpenReview:** *https://openreview.net/forum?id=ri3yPWE21Q*

## Abstract

LLM-based GUI agents have made rapid progress in understanding visual interfaces, interpreting user intentions, and executing multi-step operations across web, mobile, and desktop environments. As these agents become more capable, systematic and reproducible evaluation has become essential for measuring progress and identifying remaining weaknesses. This survey provides a comprehensive overview of benchmarks for LLM-based GUI agents, covering three major categories: grounding and QA tasks, navigation and multi-step reasoning tasks, and open-world environments that reflect realistic and dynamic software usage. We examine how existing benchmarks evaluate both component-level abilities, such as intent understanding, GUI grounding, navigation, and context tracking, and system-level abilities, such as adaptation, personalization, privacy protection, safety, and computational efficiency. By comparing datasets, environments, and evaluation metrics, the survey reveals clear trends in benchmark design, along with persistent gaps including limited adaptability, vulnerability to malicious interfaces and prompt attacks, lack of interpretability, and significant computational overhead. We highlight emerging directions such as safety-aware evaluation, human-centered evaluation, personalization, lightweight deployment, and zero-shot generalization. This survey aims to serve as a practical guide for researchers who design GUI agents, build benchmarks, or study LLM-driven user interface automation. We further provide a cross-benchmark audit that clarifies capability coverage and comparison boundaries, arguing that GUI-agent benchmark scores should be interpreted within benchmark families rather than as a single universal leaderboard.

## 1 Introduction

A graphical user interface (GUI) enables users to interact with complex digital systems through visual elements like icons and buttons. GUI agents represent an important, application-oriented branch in the field of artificial intelligence. They are designed to mimic human users by performing atomic operations such as clicks, inputs, and scrolls to interact with complex digital environments, including web browsers, mobile applications, and desktop software. GUI agents demonstrate tremendous potential in numerous

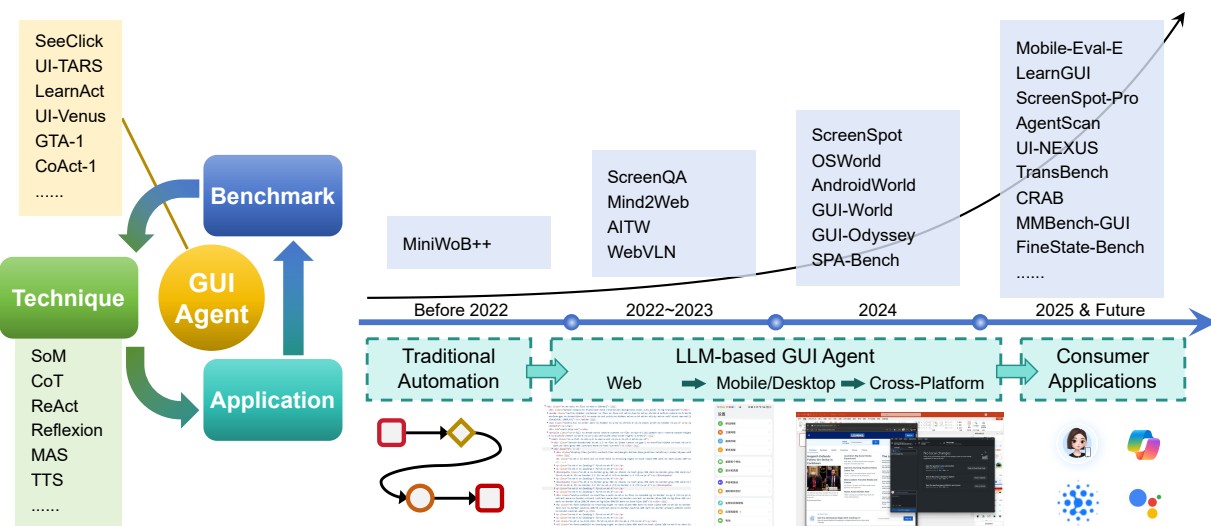

Figure 1: Co-evolution of GUI agent techniques, benchmarks, and applications.

domains, such as enterprise software automation, mobile device assistance, creative tool manipulation, and accessibility technologies. However, GUI agents face unique challenges. They must accurately identify, locate, and comprehend the function of interface elements from visually dense layouts. Furthermore, they must comprehend the high-level goals and intentions of users and effectively decompose tasks into a sequence of executable operations.

Recently, the rise of Large Language Models (LLMs), particularly Multimodal LLMs (MLLMs), has provided a powerful solution to overcome these challenges. The exceptional reasoning, knowledge integration, and natural language understanding capabilities of LLMs have been integrated into the cognitive core of GUI agents, greatly empowering their task understanding, long-term planning, and adaptation abilities. This synergy marks a significant shift in the field, moving GUI agents from rule-driven approaches toward intelligent, decision-driven systems. Advanced agents (Liu et al., 2024; Qin et al., 2025; Song et al., 2025; Gu et al., 2025) have shown impressive performance in understanding screen content and executing appropriate actions.

Given the significant leap in capabilities achieved by LLM-based GUI agents, a systematic, fair, and reproducible evaluation of their core competencies has become crucial for measuring progress and advancing the field. As illustrated in Figure 1, the field exhibits a dynamic co-evolution of GUI agent techniques, benchmarks, and applications. This feedback loop drives rapid progress: emerging techniques empower new applications, which in turn necessitate more challenging benchmarks to measure capability boundaries.

With new datasets and environments constantly emerging, these benchmarks exhibit substantial heterogeneity across task types, environment fidelity, and evaluation metrics, resulting in fragmentation of evaluation standards. Moreover, as the technology advances, benchmarks are shifting from simple UI grounding and QA tasks to multi-turn navigation tasks and, more recently, to open-world tasks that assess agents in authentic environments. A systematic review is essential to trace this development trajectory, distilling the trends of increasing evaluative difficulty and complexity to guide future research toward more realistic tasks. We argue that the heterogeneity among benchmarks is structural rather than incidental. While grounding benchmarks remain similar to traditional computer vision, where many methods can often be compared on a small number of widely recognized datasets, navigation and open-world benchmarks increasingly resemble reinforcement-learning-style sequential decision problems, where observability, action spaces, environment dynamics, and evaluator design materially affect what capability is being measured. Consequently, no single benchmark can plausibly cover the full spectrum of agent ability, and benchmark scores must be interpreted family by family rather than as one universal leaderboard. In this survey, we aim to systematically review existing benchmarks for GUI agents, establishing a clear taxonomy of task scenarios, encompassing grounding & QA,

navigation tasks, and open-world environments. By comparing benchmark designs across datasets, environments, and evaluation metrics, we distill prominent trends in the evolution of GUI agent assessment and identify current limitations of agents and benchmarks, such as limited adaptability, vulnerability to malicious attacks, lack of explainability evaluation, and considerable computational cost. Furthermore, we highlight emerging research directions deserving closer attention, including safety-aware evaluation, human-centered evaluation, personalization, lightweight deployment, and zero-shot generalization. Our contributions are:

- *Evaluation-centric taxonomy.* We organize GUI-agent benchmarks into three benchmark families: GUI Grounding and QA, Navigation and Multi-Step Reasoning, and Open-World Environments.

- *Benchmark architecture analysis.* We analyze benchmarks through three structural pillars: dataset, environment, and evaluator, making differences in task coverage, environment fidelity, and metrics easier to interpret.

- *Cross-benchmark audit.* We provide a systematic audit of capability coverage and comparison boundaries, showing why benchmark scores should often be interpreted family by family rather than as a single global ranking.

- *Design roadmap.* We identify under-evaluated dimensions, including explainability, generalization, safety and privacy, human-centered evaluation, personalization, and lightweight deployment, and discuss their implications for future benchmark design.

**Difference with Existing Surveys**   While several recent surveys have examined the landscape of GUI agents, they mostly focus on agent architectures and methodological techniques, treating benchmarks as one component within a broader review. For instance, (Nguyen et al., 2025) and (Zhang et al., 2025) provide comprehensive overviews of agent architectures, and (Shi et al., 2025) specifically concentrates on safety and trustworthiness. In contrast, we position this survey as an **evaluation-centric** complement to these method-centric surveys. More specifically, our survey differs from existing ones in three aspects. First, we organize the literature around task scenarios ranging from grounding and QA, to multi-step navigation, and finally to complex open-world environments, thereby enabling us to discuss benchmark evolution alongside the increasing realism and difficulty of GUI tasks. Second, we analyze each benchmark through the three structural pillars of dataset, environment, and evaluator, enabling a more focused comparison of task coverage, environment types, and evaluation metrics, and making benchmark differences easier to interpret. Third, beyond summarizing benchmark leaderboards, we highlight the persistent gaps between benchmark scores and real-world deployment. By focusing on the structure and evolution of evaluation methodologies, this work complements existing method-centric surveys and provides a roadmap for the next generation of rigorous GUI agent evaluation.

**Survey Scope and Methodology**   This survey focuses on publicly available benchmarks and evaluation resources for LLM-based GUI agents across web, mobile, desktop, and cross-platform settings. We collected candidate papers and benchmark resources from public sources, including arXiv, GitHub repositories, and papers published in relevant journals and conference venues. The candidate pool was further expanded through backward and forward citation tracing. We mainly consider benchmark papers published since 2023, while also including selected influential earlier works. A work is included as a core benchmark entry if it introduces a benchmark, dataset, environment, evaluator, or standardized evaluation setup directly used to assess GUI agents. Papers whose primary contribution is agent architecture or training without a distinct benchmark contribution are cited when they provide necessary context for interpreting benchmark results, but are not treated as core benchmark entries. For summary tables and case studies, we prioritize representative benchmarks that are publicly documented, influential in subsequent work, or clearly reflect a distinct task scenario, environment type, or evaluation setting. Finally, leaderboard results in case studies are used mainly as illustrative snapshots of the field rather than controlled cross-benchmark comparisons, because models, splits, update times, and evaluation settings are not fully unified across sources.

In the rest of this survey, Section 2 introduces the principles for benchmarking LLM-based GUI agents. Section 3 describes the taxonomy of existing benchmarks. Section 4 discusses the challenges and future directions of GUI agents and benchmarks. Section 5 concludes this paper.

## 2 Principles for Benchmarking

This section describes what to evaluate and how to evaluate GUI agents. We begin by outlining the general pipeline and key components of a GUI agent, followed by a discussion of the evaluation objectives. We then present the overall architecture of benchmarks.

### 2.1 General Pipeline of a GUI Agent

As shown in Figure 2, the general pipeline of an LLM-based GUI agent operates as a continuous loop encompassing four main components: the *Perceiver*, *Planner*, *Action Executor*, and *Memory*. This iterative cycle allows the agent to complete tasks by observing its environment (the GUI), deciding on a course of action, executing that action, and learning from the result. For a more detailed understanding, interested readers may refer to comprehensive surveys on GUI agents, e.g., (Nguyen et al., 2025; Shi et al., 2025; Zhang et al., 2025).

**Perceiver.** As the foundational step, GUI perception is crucial for agents and primarily relies on two main modalities: structural and visual. Structural perception parses machine-readable data, such as HTML/Document Object Model (DOM) for websites or accessibility trees (A11y trees) for mobile/desktop software. While structural data offers robust element identification and interaction, it is not always accessible. In contrast, visual perception analyzes screenshots using computer vision techniques like OCR (Qian et al., 2022) to detect buttons, text, or icons. The robust visual capabilities of MLLMs enable agents to understand GUIs across different environments.

**Planner.** LLMs equip GUI agents with hierarchical planning capabilities. At the high level, after perceiving the GUI state, the agent understands user intent and decomposes complex, multi-step tasks into a sequence of simpler subtasks. It often employs a Chain-of-Thought (CoT) (Wei et al., 2022) approach to maintain long-term consistency. At the low level, agents determine the next atomic action to execute based on the high-level plan and the current GUI observations.

**Action Executor.** This component uses tools to interact with UI elements and execute the planned action in the environment, including simulated mouse clicks, keyboard presses, and touch gestures. Common tools include the Android Debug Bridge (ADB), libraries like `PyAutoGUI`, and system-level accessibility APIs. Virtual or simulated environments, such as Android emulators (Rawles et al., 2025), are frequently employed to facilitate scalable experimentation and access to richer interaction signals.

**Memory.** This component maintains context and improves agent performance over time. Memory is typically categorized as internal/short-term (Lu et al., 2023) and external/long-term (Wang et al., 2023). Internal memory records actions, observations, and reasoning steps within a task, while external memory stores broader UI/task knowledge, including past successful trajectories. Recent work further explores reusable transition graphs as structured GUI experience, enabling mobile GUI agents to reuse past interaction trajectories for more efficient execution (Zheng et al., 2026).

### 2.2 Evaluation Objectives

To systematically assess the performance of LLM-based GUI agents, both the capabilities of individual components in the pipeline and the overall readiness for real-world deployment should be considered. Accordingly, we categorize the evaluation objectives into component-level and system-level capabilities. The former corresponds to the functional abilities embodied by the components of agents, while the latter reflects the practical demands for real-world deployment. As illustrated in Figure 2, this categorization aligns directly with the pipeline of GUI agents, providing a clear connection between agents and evaluation criteria.

**Component-Level Capabilities.** As mentioned above, component-level capabilities are important for reflecting the functions of each individual component in the GUI agent pipeline. These foundational abilities are prerequisites for successful task execution.

- *Intent Understanding.* The GUI agent must accurately recognize, parse, and ultimately comprehend the user queries and the ultimate goal of the task. Since user instructions can often be ambiguous or incomplete, the ability to clarify and disambiguate user intent is essential to ensure proper task alignment.

- *GUI Grounding.* This capability concerns the precise identification and localization of UI elements, including buttons, images, links, and text input fields. Reliable element localization forms the basis for all downstream interactive operations.

- *Navigation.* GUI agents must be able to plan and execute a sequence of actions to navigate through applications and achieve the goal of the task. This multi-step reasoning process requires the agent to deeply understand the logical dependencies and temporal relationships between individual actions.

- *Context Tracking.* Agents should be able to store and retrieve relevant information from past interactions. This memory ability is vital for supporting multi-turn interactions, allowing agents to maintain task coherence by recalling previous steps, user preferences, or data entered across multiple conversational exchanges.

**System-Level Capabilities.** Beyond core capabilities, the following system-level objectives evaluate the maturity and reliability of GUI agents for real-world deployment.

- *Adaptation.* GUIs vary across platforms, operating systems, applications, and versions. A practical agent must perform reliably in diverse interfaces it has not seen before. High adaptability is a key prerequisite for real-world deployment. Furthermore, in authentic dynamic environments, the agent must demonstrate robustness in handling unexpected events (e.g., pop-ups, advertisements) and anomalies (e.g., UI response latency, element loading failures) to ensure uninterrupted task completion.

- *Personalization.* An advanced GUI agent should possess the capacity to learn and internalize user preferences and habits. This learned information should then be used to inform and refine its action strategy. This context-aware and customized capability enables the agent to provide a more efficient and tailored interaction experience, significantly enhancing the utility of agents and user satisfaction.

- *Privacy Protection.* GUI operations frequently involve personal and sensitive information of users. Consequently, a trustworthy GUI agent must be designed with robust security and privacy protection mechanisms. This includes the inherent need to recognize and reject illicit or harmful instructions, and to effectively counter malicious safety threats encountered in the real world, such as image forgery of UI elements, malicious pop-ups, or overlays. Indirect prompt injection poses a particularly serious challenge for LLM-based agents.

- *Computational Efficiency.* It is critical for optimal user experience and system scalability that GUI agents operate within reasonable computational and temporal constraints.

### 2.3 Benchmarks Architecture

A well-designed benchmark must comprehensively capture the process through which a GUI agent operates and is evaluated. To this end, we posit that a complete agent benchmark is structured around three pillars, as illustrated in Figure 2: the dataset, the environment, and the evaluator. The dataset defines the task and provides the ground truth; the environment offers the executable interface and stateful platform on which the agent performs actions; and the evaluator measures performance under well-defined metrics. Taken together, these components establish a reproducible and holistic framework for assessing GUI agents.

**The Dataset.** The dataset provides the foundational raw materials that instantiate the task scenario. It supplies the specific instructions, contexts, and reference solutions needed for evaluation. A dataset typically includes:

- *Task Instructions*: Natural language descriptions of user intent, which can range from atomic commands to complex, multi-intent instructions.

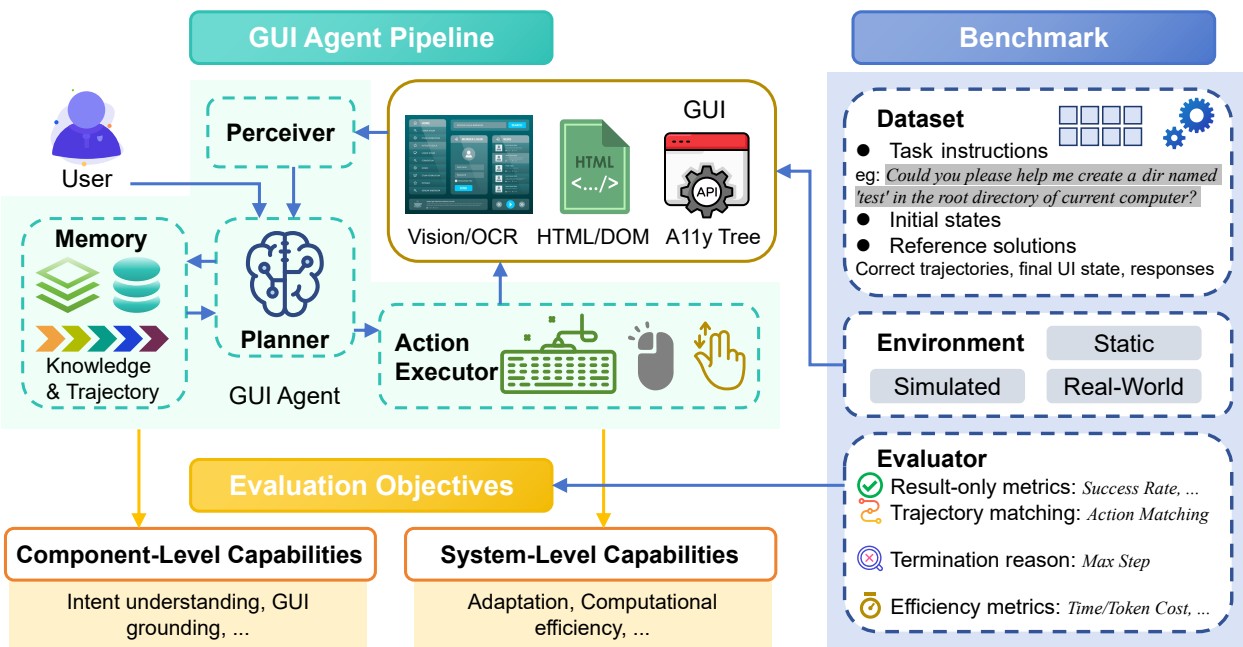

Figure 2: General pipelines and evaluation objectives of GUI agents and architecture of benchmarks.

- *Initial States*: The starting conditions of the task, provided as screenshots, DOM/accessibility trees, or a combination thereof.

- *Reference Solutions*: The ground truth information crucial for automated evaluation, which may include the correct action trajectory, the expected final state of the UI, or a canonical successful response.

To increase diversity, many modern benchmarks utilize parameterized task templates or systematic task generators to create vast and diverse sets of instructions from the dataset.

**The Environments.** The operational environments of benchmarks dictate the complexity and authenticity of the interaction. They can be grouped into three fundamental types:

- *Static Environments* (Non-Interactive): These environments capture GUI interfaces as fixed, non-interactive representations, such as screenshots or HTML snapshots. They record interactions as static data, enabling reproducible evaluation without needing the actual application to run. They are primarily used to test GUI grounding capabilities, as the actions do not change the environment state.

- *Simulated Environments* (Controlled Interaction): These environments recreate controlled, isolated, dynamic interaction scenarios, often using web page rendering engines (e.g., MiniWoB, WebArena) or simplified OS emulators. They strike a balance between reproducibility and dynamism, allowing the environment to respond to the agent's actions. This makes them well suited for scalable experimentation and training.

- *Real-World Environments* (Authentic Testing): Tasks run directly on live applications or websites, often utilizing tools like the Android Debug Bridge (ADB) or desktop automation libraries. These benchmarks offer the most authentic, comprehensive, and dynamic testing ground, directly mirroring real-world applicability. However, they present significant challenges in terms of reproducibility, execution stability, and handling out-of-distribution events.

**The Evaluators.** The evaluators are the scoring components that assess the performance and abilities of GUI agents based on various metrics and measurements. Metrics are the quantitative standards used to formalize this assessment. Metrics can be categorized by their evaluation methodology and the objectives they measure. Common evaluation methods and metrics include:

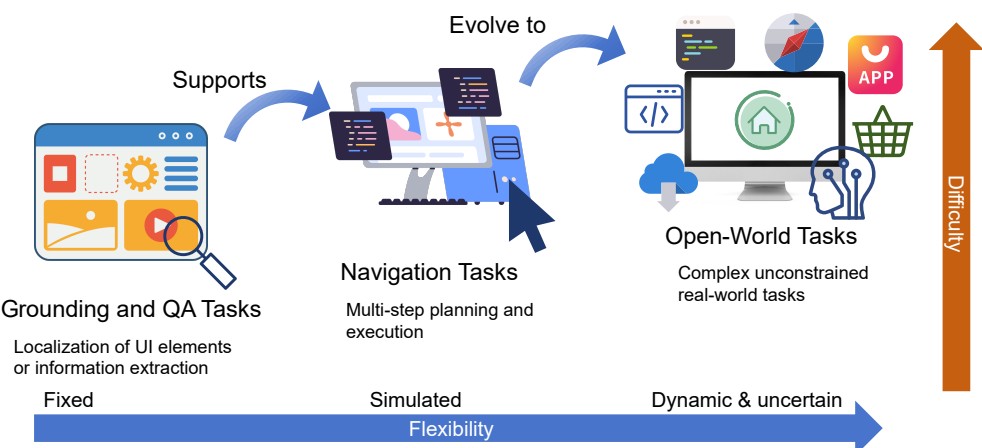

Figure 3: Task scenarios of GUI agent benchmarks.

- *Result-Only Metrics.* These metrics focus solely on the final outcome of the task. *Accuracy* and *Success Rate* are typical examples, measuring the grounding and navigation abilities of GUI agents, respectively. *Intersection over Union (IoU)* is a fine-grained measure of localization precision, which calculates the ratio of the intersection to the union of the predicted bounding box and the ground-truth box.

- *Trajectory Matching.* For more fine-grained evaluation of GUI agents, benchmarks measure the similarity or exact match between the predicted action sequence of the agent and the ground-truth trajectory. *Action Matching Score* is a representative metric. For a step to be successful, the action type (e.g., click, text input), the target UI element, and the action parameters must all be correct.

- *Termination Reason.* These metrics help researchers understand exactly why an agent failed or succeeded. The termination can be classified into several key categories: (1) Self-Reported Completion (SRC): the agent autonomously ends the task. It can be further divided into true success or premature termination, where the latter means that the agent reports success while the task was not actually finished. (2) Maximum Step Reached (MSR): the execution has exceeded a predefined maximum number of steps. A subcategory is Overdue Termination, in which MSR occurs because the agent has effectively completed the task but failed to recognize its completion. (3) Error & Collapse. (4) Deemed Impossible: the agent proactively determines that the task is impossible to complete based on the current environment or constraints. This determination can be accurate or erroneous.

- *Efficiency Metrics.* Efficiency is typically quantified by *Time Cost (per Step)* and *Token Cost (per Step)*, which directly represent the computational and monetary resources consumed by the underlying LLMs for generation and processing.

- *Reliability Metrics.* These metrics guide GUI agents toward reliability and trustworthiness. *Accuracy/SR variation in different environments* intuitively measures the adaptability of a GUI agent. The *Harm Prevention Rate* indicates the proportion of risky tasks in which GUI agents can successfully avoid negative outcomes for users, such as privacy leaks, economic losses, and ethical violations in real-world scenarios.

## 3 Taxonomy of Existing Benchmarks

This section provides an overview of GUI agent benchmarks. The taxonomy is fundamentally driven by the task scenarios that benchmarks are designed to address. The task scenario defines the core intellectual challenges posed to agents, inherently reflecting the evolution of required agent functions and goal complexity. We observe that progress in GUI agent benchmarking has proceeded from focusing on single-step GUI grounding & QA tasks to multi-step navigation tasks, and more recently to challenging open-world tasks.

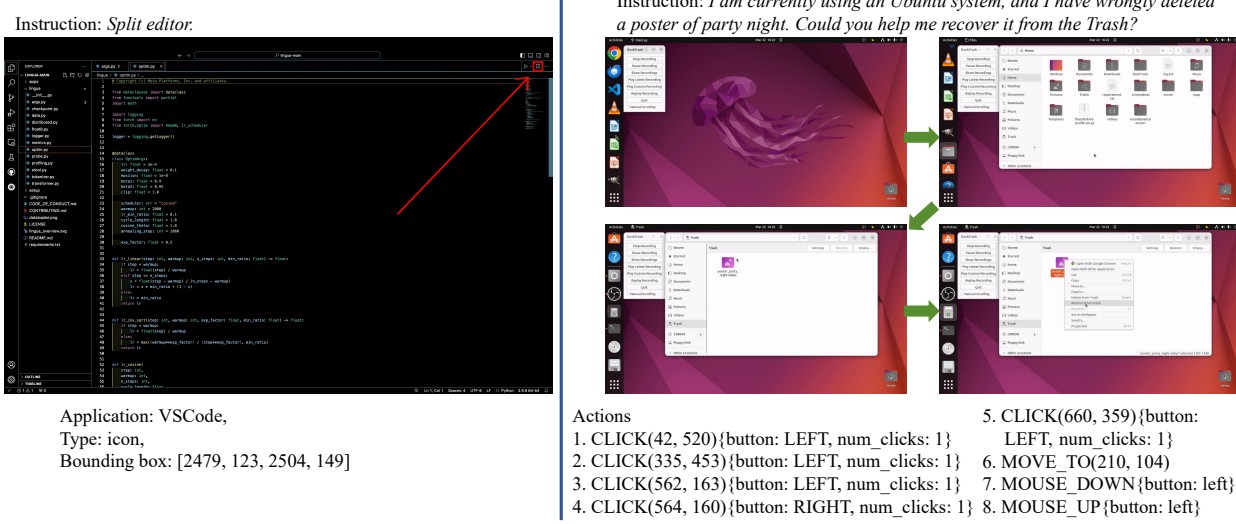

Instruction: *Split editor.*

Application: VSCode,
Type: icon,
Bounding box: [2479, 123, 2504, 149]

Instruction: *I am currently using an Ubuntu system, and I have wrongly deleted a poster of party night. Could you help me recover it from the Trash?*

Actions
1. CLICK(42, 520){button: LEFT, num_clicks: 1}
2. CLICK(335, 453){button: LEFT, num_clicks: 1}
3. CLICK(562, 163){button: LEFT, num_clicks: 1}
4. CLICK(564, 160){button: RIGHT, num_clicks: 1}
5. CLICK(660, 359){button: LEFT, num_clicks: 1}
6. MOVE_TO(210, 104)
7. MOUSE_DOWN{button: left}
8. MOUSE_UP{button: left}

Figure 4: Examples of GUI grounding tasks in ScreenSpot-Pro (left) and open-world tasks in OSWorld (right). Note that open-world tasks serve as a comprehensive example, representing the advanced evolution of navigation tasks in authentic environments.

- **GUI Grounding & QA Tasks** demand precise localization of UI elements or information extraction from the GUI based on a natural language query, primarily testing the basic GUI perception and understanding capabilities of agents.

- **Navigation Tasks** challenge an agent to plan and orchestrate a sequence of executable actions to fulfill the goal of the user, focusing on its multi-step planning and sequential action execution.

- **Open-World Tasks** evaluate the agent by deploying it in authentic, dynamic, and often cross-application workflows that embody the inherent diversity and unpredictability of real-world software environments, serving as a comprehensive test of the integrated component-level and system-level capabilities.

Grounding & QA is the necessary prerequisite for navigation, while open-world tasks represent the evolution and generalization of navigation tasks. The relation of these task scenarios is illustrated in Figure 3. To demonstrate the task spectrum, Figure 4 visualizes two representative examples: the fundamental GUI grounding tasks and the complex open-world tasks. In what follows, we will systematically review the evolution of these benchmarks and delve into the representative methods and performance. Crucially, organizing the taxonomy by task scenario is the optimal approach for tracking the co-evolution of agent capabilities and evaluation standards, as it clearly reveals how the rigor of assessment is constantly elevated across datasets, environments, and evaluators.

## 3.1 GUI Grounding and QA Benchmarks

As introduced above, the first task type is primarily concerned with addressing the fundamental problem of understanding and perception.

### 3.1.1 Overall Comparison

Typical benchmarks include ScreenQA (Hsiao et al., 2025) and ScreenSpot (Cheng et al., 2024). ScreenQA requires the agent to answer questions or extract information from screenshots and metrics like Average F1 score are applied. ScreenSpot measures the grounding accuracy of GUI agents with natural language instructions. The subsequent versions of ScreenQA are ScreenQA-Short and Complex-ScreenQA (Baechler et al., 2024), which introduced alternative short answers and more challenging questions. An overview of

Table 1: Benchmarks focusing on grounding and QA tasks.

| Benchmarks | Dataset | Environments | Evaluators | Venue | URL |
|---|---|---|---|---|---|
| ScreenQA | Based on RICO dataset, 86k+ QA pairs | Static Images (Mobile) | F1 Score | NAACL 2025 | GitHub |
| ScreenSpot | Artificially cropped screenshots | Static Images (Cross-platforms) | Accuracy | ACL 2024 | Hugging Face |
| ScreenQA-Short | Based on ScreenQA with alternative short answers | Static Images (Mobile) | F1 Score | IJCAI 2024 | GitHub |
| Complex-ScreenQA | Extension to ScreenQA-Short focusing on reasoning | Static Images (Mobile) | F1 Score | IJCAI 2024 | GitHub |
| GUI-World | 12k videos, 100k QA pairs | Dynamic GUI (Cross-platforms) | QA Performance (LLM-as-a-Judge) | ICLR 2025 | GitHub |
| ScreenSpot-V2 | Based on ScreenSpot, 1.2k+ revised grounding instructions | Static Images (Cross-platforms) | Accuracy, IoU | arXiv | Hugging Face |
| ScreenSpot-Pro | 1.5k+ hi-res professional application screenshots | Static Images (Desktop) | Accuracy | ICLR 2025 | GitHub |
| OSWorld-G | Based on OSWorld, 564 fine-grained annotated samples | Real OS (Desktop) | Accuracy, Success Rate | arXiv | GitHub |
| TransBench | 1.4k+ cross-platform/version screenshots, 22k+ instructions | Static Images (Cross-platforms) | Accuracy, Average Distance | ACL 2025 | TransBench |

grounding and QA benchmarks is given in Table 1. We can observe that benchmarks in this domain are evolving to include more complex and realistic tasks, as well as more comprehensive metrics, across diverse data and environments. Shifting from static snapshots to dynamic trajectories, GUI-World (Chen et al., 2025b) includes videos and key frames to evaluate the dynamic GUI-understanding ability of agents across a wide spectrum of environments ranging from desktop and mobile to extended reality (XR). Furthermore, GUI-World introduced the LLM-as-a-Judge methodology to assign a similarity score between the response of agents and the reference answer. ScreenSpot-V2 (Wu et al., 2024) fixed problems in ScreenSpot and introduced the Intersection over Union (IoU) metric to more precisely evaluate the quality of agents in locating UI elements. ScreenSpot-Pro (Li et al., 2025) is a recent advancement of ScreenSpot that focuses on high-resolution computer screens and professional scenarios. OSWorld-G (Xie et al., 2025) extends this scope of simple element localization to include fine-grained operations (e.g., character-level cursor positioning, slider control) and rejection handling for infeasible commands (e.g., opening non-existent software). TransBench (Lu et al., 2025b) evaluates the adaptability of GUI agents through grounding tasks across different platforms, applications, and versions.

### 3.1.2 Case Study

As the prerequisite for any effective operation, the UI grounding task serves as a foundational test for all contemporary GUI agents, irrespective of their architectural backbones (LLMs or MLLMs). To address these tasks, the evolution of agent technologies can be broadly categorized into three principal paradigms based on their perceptual input strategy:

- *Structural Text.* Early LLM-based agents relied exclusively on parsing GUI interfaces into structured textual representations, such as HTML/DOM or accessibility trees. The role of LLMs is to process this complex text alongside the user instruction. However, methods based solely on structure remain fragile when facing inaccessible structural data or dynamic content.

- *Visual Perception.* These limitations led to the integration of visual perception, enabled by the recent emergence of Vision-Language Models (VLMs). A pivotal technique in this domain is Set-of-Marks prompting (Yang et al., 2023), which involves augmenting screenshots with visual markers, typically numbered or lettered bounding boxes, to clearly delineate interactive components. This marking process

Table 2: The grounding accuracy (%) leaderboard of ScreenSpot-Pro.

| Model | Applications | | | | | | AVG perf. |
|---|---|---|---|---|---|---|---|
| | Development | Creative | CAD | Scientific | Office | OS | |
| Hcompany/Holo2-30B-A3B | 69.9 | 61.0 | 54.8 | 64.6 | 81.7 | 67.9 | 66.1 |
| GTA1-32B | 61.2 | 52.8 | 60.5 | 65.0 | 83.5 | 65.3 | 63.6 |
| Holo1.5-72B | 63.5 | 62.5 | 51.3 | 64.2 | 79.6 | 59.7 | 63.3 |
| UI-Venus-72B | 59.5 | 55.4 | 57.5 | 66.5 | 77.8 | 57.7 | 61.9 |
| UI-TARS-1.5 | 63.9 | 50.4 | 58.2 | 69.3 | 79.6 | 51.0 | 61.6 |
| Seed-1.5-VL | 53.8 | 59.2 | 59.0 | 61.4 | 74.8 | 60.2 | 60.9 |
| GUI-ARP-7B | 59.2 | 52.5 | 61.7 | 62.2 | 77.8 | 54.6 | 60.8 |
| GUI-AIMA-3B | 55.2 | 48.7 | 57.5 | 64.2 | 79.6 | 58.7 | 59.6 |
| Hcompany/Holo2-8B | 55.5 | 52.2 | 52.5 | 61.0 | 80.0 | 56.6 | 58.9 |
| GTA1-Qwen2.5VL-72B | 57.2 | 51.0 | 49.8 | 63.0 | 77.0 | 57.1 | 58.4 |

often utilizes specialized element detection models for precise component delineation and Optical Character Recognition to capture text elements. The VLM then processes the visually augmented screenshot.

- *Comprehensive Multimodal Modeling.* Some advanced GUI agents have shifted towards a comprehensive multimodal modeling paradigm, recognizing that optimal grounding performance requires synthesizing information from all available data sources to overcome the limitations of any single modality. Supervised Fine-Tuning (SFT) and Reinforcement Learning (RL) on high-quality multimodal data help agents achieve better performance.

To give readers a clear view of the current performance landscape of GUI agents, we report results on ScreenSpot-Pro, which evaluates grounding performance on professional computer applications. The leaderboard is shown in Table 2. As can be seen, high-resolution screens and professional software pose significant challenges for agents, and even the current state-of-the-art (SOTA) performance is far from satisfactory, revealing that although GUI grounding appears to be a relatively mature basic task, perception remains a major bottleneck in complex desktop environments. Recent multimodal agents, Holo2 (Andreux et al., 2025), GTA1 (Yang et al., 2025c), and UI-Venus (Gu et al., 2025) demonstrate leading performance. Their strong performance is typically achieved by leveraging SFT and RL on high-fidelity multimodal inputs, encompassing both screenshots and underlying structural HTML/A11y tree data.

## 3.2 Navigation and Multi-Step Reasoning Benchmarks

As the capabilities in single-step grounding tasks and intent understanding improved, the research focus shifted to navigation tasks. These benchmarks assess the planning and multi-step reasoning abilities of GUI agents and require them not only to perceive but also to execute the correct sequence of operations.

### 3.2.1 Overall Comparison

Table 3 summarizes the key navigation benchmarks. MiniWoB++ (Liu et al., 2018) served as a pioneering benchmark focusing on simple web interaction tasks evaluated primarily by task success rate. However, the rapid advancement of LLMs has enabled modern agents to achieve near human-level performance on these simple tasks. WebArena (Zhou et al., 2024) was another notable benchmark for web GUI agents. It includes tasks that are intentionally impossible to complete, marking an early exploration of self-awareness in agents.

A pivotal development in navigation evaluation is the adoption of more fine-grained metrics alongside the expansion of dataset scales. Mind2Web (Deng et al., 2023) utilized static real-world traces and introduced metrics such as Element Accuracy, Operation F1, and Step Success Rate to measure the correctness of the entire action trajectory. This methodology was extended to the mobile domain by AITW (Rawles et al., 2023), AndroidControl (Li et al., 2024), and GUI-Odyssey (Lu et al., 2025a), which employed Action Matching Scores and Step-wise Accuracy to assess operation precision. OmniAct (Kapoor et al., 2024), UI-Vision (Nayak et al., 2025), and AgentNetBench from OpenCUA (Wang et al., 2025b) further generalized this fine-grained evaluation approach to desktop and cross-platform environments. Simultaneously, the

Table 3: Benchmarks focusing on navigation tasks.

| Benchmarks | Dataset | Environments | Evaluators | Venue | URL |
|---|---|---|---|---|---|
| MiniWoB++ | Based on MiniWoB, 104 web interaction tasks | Simulated (Web) | Success Rate (SR) | NeurIPS 2018 | GitHub |
| Mind2Web | 2350 tasks from 137 websites | Static Real Traces (Web) | Element Accuracy, Operation F1, Step SR | NeurIPS 2023 | GitHub |
| AITW | 715k episodes spanning 30k unique instructions | Static Real Traces (Mobile) | Action Matching Scores | NeurIPS 2023 | GitHub |
| WebArena | 812 tasks derived from 241 templates | Simulated (Web) | SR | ICLR 2024 | GitHub |
| WebVLN | 8990 records/paths with 14825 QA pairs derived from 3 shopping websites | Simulated (Web) | SR, SR weighted by Path Length | AAAI 2024 | GitHub |
| VisualWebArena | Based on WebArena, 910 new visual web tasks | Simulated (Web) | Success Rate | ACL 2024 | GitHub |
| MT-Mind2Web | 720 multi-turn tasks based on Mind2Web | Simulated (Web) | Element Accuracy, Operation F1, Step/Turn SR | ACL 2024 | Hugging Face |
| OmniAct | 9802 task instructions, screenshots, and ground truth `PyAutoGUI` scripts | Static Real Traces (Cross-platforms) | Action Score, Sequence Score | ECCV 2024 | Hugging Face |
| AndroidControl | 15283 unique tasks over 833 Android apps | Simulated (Mobile) | Step-wise Accuracy | NeurIPS 2024 | GitHub |
| MobileAgentBench | 100 tasks across 10 open-source apps | Simulated (Mobile) | SR, Termination Reason, Time/Token Cost | AAAI 2025 | GitHub |
| GUI-Odyssey | 8334 episodes spanning 6 types of cross-app tasks, 212 apps, and 1.4k app combos | Simulated (Mobile) | SR, Action Matching Score | ICCV 2025 | GitHub |
| VideoWebArena | 2021 tasks based on video tutorials | Simulated (Web) | SR, Intermediate Score, Final Score | ICLR 2025 | GitHub |
| UI-Vision | 8227 tasks on 83 apps with dense annotations | Simulated (Desktop) | Step Success Rate | arXiv 2025 | GitHub |
| AgentNetBench | 100 representative desktop tasks derived from 22.6k trajectories | Static Real Traces (Desktop) | Step SR | NeurIPS 2025 | GitHub |
| FineState-Bench | 2257 fine-grained tasks | Simulated (Cross-platforms) | Locate SR, Interact SR | arXiv 2025 | GitHub |

field witnessed a significant increase in data scale, evolving from hundreds of tasks in early benchmarks to thousands of trajectories, which is essential for training and evaluating robust GUI agents.

Beyond basic Success Rate (SR) and Step-wise SR, more advanced metrics have been proposed to capture the efficiency and reliability of GUI agents. WebVLN (Chen et al., 2024) introduced efficiency-oriented metrics, such as SR weighted by Path Length, to penalize redundant actions. To assess visual robustness, VisualWebArena (Koh et al., 2024) extended the WebArena environment with 910 visually intensive tasks. VideoWebArena (Jang et al., 2025) further expanded the modality to video understanding, creating tasks based on video tutorials and proposing Intermediate and Final Scores to measure performance. In terms of interaction complexity, MT-Mind2Web (Deng et al., 2024) generalized navigation to multi-turn scenarios, requiring agents to handle conversational interactions and maintain context over long horizons. Aiming for better interpretability and precision, MobileAgentBench (Wang et al., 2025a) introduced Termination Reasons that categorize failures into specific types to facilitate detailed error diagnosis. FineState-Bench (Ji et al., 2025) focused on fine-grained state control to test the ability of agents to handle precise operations.

### 3.2.2 Case Study

To interpret performance on navigation benchmarks, it is useful to briefly discuss advanced alignment and training paradigms that are widely used in GUI agents. As discussed in Section 3.1.2, approaches such as SFT and RL are also widely used in navigation tasks.

- *Supervised Fine-Tuning (SFT).* SFT helps agents learn fundamental UI interaction patterns, accurate element grounding, and the mapping from natural language instructions to executable low-level actions. With high-quality human demonstration data, SFT is crucial for achieving high fidelity in element identification and action prediction, thereby providing a robust base for subsequent, more advanced training. Recent efforts such as OpenCUA (Wang et al., 2025b) further show that scaling cross-OS demonstrations, compact state-action trajectories, and reflective long CoT supervision within an SFT pipeline can substantially strengthen computer-use agents.

- *Reinforcement Learning (RL).* Guided by either environmental success signals or explicit rule-based rewards, RL or Reinforcement Fine-Tuning (RFT) optimize agent policies to maximize long-term cumulative reward, thereby achieving deeper alignment with complex, multi-step task objectives. Advanced RL algorithms, such as Direct Preference Optimization or Group Relative Policy Optimization, can equip agents with stronger adaptation to unseen UI layouts and greater robustness through self-correction when encountering errors. Consequently, advanced GUI agents predominantly rely on a two-stage training paradigm (SFT → RL), or incorporate multi-turn RL techniques to reach SOTA performance.

In Table 4, we list representative performance results on GUI-Odyssey, whose long-horizon cross-app navigation tasks have an average of 15.4 steps. This benchmark evaluates GUI agents using two core metrics, Action Matching Score (AMS) and Success Rate (SR). The GUI-Odyssey performance data clearly illustrate the current state of agent development. General-purpose MLLMs, such as GPT-4o and Claude-computer-use, exhibit very low SRs of 5.4% and 3.1%, respectively. In contrast, recent GUI agents like UI-Venus and UI-TARS (Qin et al., 2025) raise SR to over 70% and, in some cases, close to 90%. Their strong results are attributed to their end-to-end native agent architectures. Furthermore, these agents leverage large-scale training and often incorporate RL. However, this success is still heavily reliant on domain-specific training data. To enable agents to operate in real-world scenarios where task

Table 4: Action Matching Score (AMS, %) and Success Rate (SR, %) performance of GUI agents in GUI-Odyssey. Data collected from (Wu et al., 2024; Qin et al., 2025; Lu et al., 2025c).

| Models | AMS | SR |
|---|---|---|
| *Closed Models* | | |
| GPT-4o | 37.5 | 5.4 |
| Claude-computer-use | 60.9 | 3.1 |
| SeeClick | 71.0 | 53.9 |
| *Open Models 7B* | | |
| Qwen2.5-VL-7B | 67.4 | 52.4 |
| OS-Atlas-7B | 84.5 | 62.0 |
| UI-TARS-7B | 94.6 | 87.0 |
| UI-Venus-7B | 87.3 | 71.5 |
| UI-S1-7B | 76.3 | 59.5 |
| *Open Models 72B* | | |
| UI-TARS-72B | 95.4 | 88.6 |
| UI-Venus-72B | 87.2 | 72.4 |

types and applications exhibit combinatorial complexity, research must move beyond closed benchmarks and embrace an open-world evaluation paradigm that demands stronger reasoning and adaptation capabilities.

### 3.3 Open-World Environments and Generalization Benchmarks

Open-world tasks represent the newest development in GUI agent benchmarks, aiming to evaluate the adaptability and reliability of agents in authentic, complex, and unconstrained GUI environments. These tasks require agents to operate across applications in dynamic, unpredictable workflows and resist security risks, reflecting the inherent diversity and unpredictability of real-world software scenarios. As shown in Figure 3, open-world tasks represent an evolution and generalization of navigation tasks and currently pose the greatest challenge for assessing the practical deployment readiness of GUI agents.

### 3.3.1 Overall Comparison

OSWorld (Xie et al., 2024) and AndroidWorld (Rawles et al., 2025) are pioneering and representative examples of this transition, as they introduced scalable, realistic evaluations in desktop and mobile environments, respectively. The framework of OSWorld was later extended by WindowsAgentArena (Bonatti et al., 2024) and macOSWorld (Yang et al., 2025a) to specifically evaluate GUI agents in Windows and macOS environments. To overcome the limitation of OSWorld that tasks have to be deterministic, OSUniverse (Davydova et al., 2025) supports non-deterministic tasks, such as looking up the weather, where the result changes dynamically with the environment, requiring agents to handle non-fixed ground truth. WorldGUI (Zhao et al., 2025) embraces various initial states to better evaluate the adaptability of GUI agents in scenarios that are closer to real user settings.

Open-world benchmarks give researchers a more comprehensive understanding of the capabilities and limitations of GUI agents. With more fine-grained metrics, they also highlight the significant gap between current technology and human-level performance.

Termination reasons have been analyzed by more benchmarks recently, such as CRAB (Xu et al., 2025), SPA-Bench (Chen et al., 2025c), Mobile-Eval-E (Wang et al., 2025d), and UI-NEXUS (Guo et al., 2025b). As introduced in Section 2.3, these benchmarks categorize termination reasons into several types, including Successful, Premature, Max Steps Reached, and Collapse. This categorization helps diagnose specific weaknesses of GUI agents and provides insights for targeted improvements.

Efficiency is also considered by more benchmarks. CRAB, UI-NEXUS, and FingerTip (Yang et al., 2025b) introduced Time/Token Cost to evaluate the computational efficiency of GUI agents. MMBench-GUI (Wang et al., 2025c) also utilized Efficiency-Quality-Aware (EQA) to jointly evaluate success rate and efficiency, assigning higher scores to GUI agents that can complete tasks in fewer steps.

Recent benchmarks further push open-world evaluation toward deployment-specific weaknesses that are not fully captured. MemGUI-Bench (Liu et al., 2026b) targets memory-intensive mobile tasks that require cross-temporal or cross-spatial retention, DynamicGUIBench (Liu et al., 2026a) stresses high-dynamic desktop interfaces where single screenshots can miss important state changes, and AndroidDaily (Sui et al., 2026) studies verifiable evaluation on real-world closed-source Android applications through the GRADE diagnostic evaluator. These updates connect the open-world family more directly to long-term context, temporal observability, and process-level evaluation.

Meanwhile, benchmarks have begun to incorporate security considerations. ST-WebAgentBench (Levy et al., 2025) requires agents to complete tasks in accordance with organizational and user policies, thereby enhancing their controllability. MobileSafetyBench (Lee et al., 2024) designs tasks that involve five types of risks, especially Indirect Prompt Injection, to evaluate the harm-prevention ability of GUI agents. AgentScan (Wu et al., 2025) and macOSWorld include malicious attacks in their tasks.

Finally, LearnGUI (Liu et al., 2025) assessed the learning capability of mobile GUI agents by providing rich $k$-shot task combinations to encourage efficient knowledge transfer and generalization efficacy. FingerTip (Yang et al., 2025b) introduced two tracks, Proactive Suggestions and Personalized Execution, marking a shift of GUI agent evaluation from task completion to a higher-level assessment of how to better serve users. However, such user-oriented benchmark signals are still only partial proxies for human utility. Recent work (Chen et al., 2025a) argues that trustworthy GUI agent evaluation should also consider human oversight and privacy awareness beyond automated benchmark scores.

Through this comprehensive review, the characteristics and evolution of open-world benchmarks be-

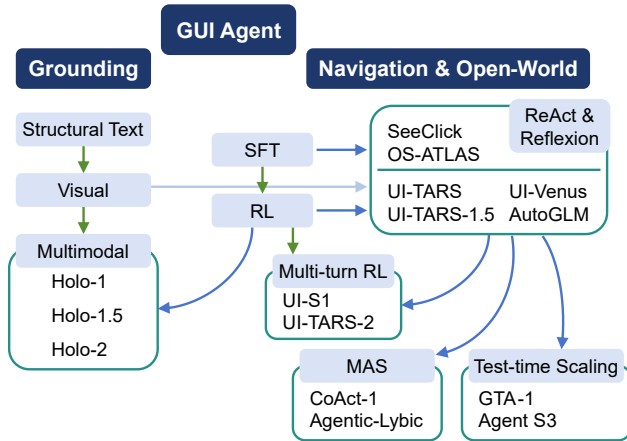

Figure 5: GUI agent technology evolution.

Table 5: Benchmarks focusing on open-world tasks.

| Benchmarks | Dataset | Environments | Evaluators | Venue | URL |
|---|---|---|---|---|---|
| OSWorld | 369 tasks on real desktop apps | Real OS (Desktop) | Success Rate (SR) | NeurIPS 2024 | GitHub |
| AndroidWorld | 116 parameterized tasks across 20 apps | Real OS (Mobile) | SR | ICLR 2025 | GitHub |
| CRAB | Generating tasks from sub-tasks | Real OS (Cross-platforms) | SR, Termination Reason, Time/Token Cost | ACL 2025 | GitHub |
| Win-dowsAgentArena | Based on OSWorld, 150+ Windows-specific tasks | Real OS (Windows) | SR | arXiv 2024 | GitHub |
| ST-WebAgentBench | 222 tasks paired with safety and trustworthiness policies and concise rules | Real OS (Web) | Completion Under Policy | ICML 2025 | GitHub |
| SPA-Bench | 340 tasks, including 150 single-app tasks and 20 cross-app tasks | Real OS (Mobile) | SR, Termination Reason | ICLR 2025 | GitHub |
| MobileSafety-Bench | 80 tasks grounded in daily life, including 5 types of risks | Real OS (Mobile) | Goal Achievement Rates, Harm Prevention Rates | arXiv 2024 | GitHub |
| Mobile-Eval-E | 25 mobile tasks including 19 multi-app tasks | Real OS (Mobile) | SR, Termination Reason, Satisfaction Score | arXiv 2025 | GitHub |
| WorldGUI | 611 tasks with various initial states | Real OS (Cross-platforms) | SR | arXiv 2025 | GitHub |
| LearnGUI | Based on AMEX and AndroidWorld, 2252 offline few-shot tasks and 101 online tasks | Real OS (Mobile) | Action Type Accuracy, Action Match Accuracy | arXiv 2025 | GitHub |
| OSUniverse | Including non-deterministic tasks | Real OS (Desktop) | SR, Weighted Score | arXiv 2025 | GitHub |
| AgentScan | Tasks with malicious instructions and glitch tokens | Real OS (Mobile) | Attack Success Rate | arXiv 2025 | arXiv |
| macOSWorld | Based on OSWorld, 202 multilingual macOS-specific tasks | Real OS (macOS) | SR, Safety Evaluation | arXiv 2025 | Website |
| UI-NEXUS | 100 task templates on 50 apps | Real OS (Mobile) | SR, Termination Reason, Time/Token Cost | arXiv 2025 | GitHub |
| MMBench-GUI | 8000+ tasks | Real OS (Cross-platforms) | SR, Efficiency-Quality-Aware (EQA) Score | arXiv 2025 | GitHub |
| FingerTip | 21437 real episodes covering 506 apps | Real OS (Mobile) | SR, Similarity, Time/Token Cost | arXiv 2025 | arXiv |
| MemGUI-Bench | 115 memory-intensive tasks and 13 standard tasks across 26 apps | Real OS (Mobile) | Multi-Attempt SR, Memory Assessment, Efficiency | arXiv 2026 | GitHub |
| Dynam-icGUIBench | 149 online tasks across 10 dynamic application domains | Real OS (Desktop) | SR | arXiv 2026 | arXiv |
| AndroidDaily | 350 daily-use tasks across 94 closed-source apps | Real OS (Mobile) | SR, Time Cost | arXiv 2026 | arXiv |

come evident: they feature more realistic environments, incorporate more complex tasks, utilize more fine-grained and diagnostic metrics, and increasingly emphasize the practicality of the agent.

### 3.3.2 Case Study

The complexity and realism of open-world benchmarks call for representative agent paradigms. This helps explain why these benchmarks matter: they expose capability bottlenecks in GUI agents and, in turn, stimulate the development of more advanced techniques. In this sense, more challenging benchmarks and advances in GUI agent technology evolve in a mutually reinforcing manner, as summarized in Figure 5.

Table 6: The Success Rate (%) leaderboard of OSWorld.

| Models | Chrome (46) | GIMP (26) | Calc (47) | Impress (47) | Writer (23) | OS (24) | Thunderbird (15) | VLC (17) | VS Code (23) | Workflow (101) | Total (369) |
|---|---|---|---|---|---|---|---|---|---|---|---|
| Agent S3 w/ GPT5 | 69.5 | 69.2 | 87.2 | 63.7 | 78.1 | 79.2 | 80.0 | 52.1 | 82.6 | 58.7 | 69.9 |
| GTA1 w/ GPT5 | 58.6 | 76.9 | 63.8 | 65.5 | 60.7 | 79.2 | 80.0 | 57.9 | 82.6 | 50.9 | 63.4 |
| Claude Sonnet 4.5 | 63.0 | 53.8 | 72.3 | 68.0 | 82.5 | 70.8 | 60.0 | 58.2 | 73.9 | 49.5 | 62.9 |
| Agentic Lybic | 63.0 | 73.1 | 60.9 | 69.4 | 65.2 | 75.0 | 66.7 | 80.7 | 82.6 | 41.6 | 61.9 |
| CoACT-1 | 54.3 | 65.4 | 70.2 | 50.3 | 73.9 | 75.0 | 73.3 | 71.9 | 78.3 | 47.9 | 60.8 |
| UI-TARS-2 | 63.0 | 50.0 | 66.0 | 56.4 | 60.9 | 41.7 | 73.3 | 49.9 | 73.9 | 34.1 | 53.1 |
| AutoGLM-OS-9B | 39.1 | 61.5 | 58.7 | 29.7 | 52.2 | 75.0 | 80.0 | 64.6 | 82.6 | 27.9 | 48.0 |
| OpenCUA-72B | 52.1 | 74.4 | 35.5 | 48.2 | 56.5 | 61.1 | 57.8 | 37.3 | 70.6 | 22.2 | 45.0 |

*ReAct.* ReAct (Yao et al., 2023) interleaves reasoning traces (*Thought*) with task-specific actions (*Action*) in an observe-think-act loop, enabling synergistic planning and execution across multi-step decision tasks.

*Reflexion.* Reflexion (Shinn et al., 2023) employs a meta-cognitive feedback loop where the agent self-critiques failed trajectories to generate refined strategies, storing them in memory for robust error recovery.

*Multi-Agent System (MAS).* Multi-Agent Systems (Hong et al., 2024) distribute complex workflows across multiple specialized agents, improving scalability and fault tolerance through hierarchical organization and parallel execution.

*Test-Time Scaling.* Test-Time Scaling (Yang et al., 2025c; Gonzalez-Pumariega et al., 2025) enhances inference by sampling multiple action proposals or entire trajectories, significantly improving decision quality and overall success rates.

In Table 6, we present the representative performance data on OSWorld. The leaderboard provides a breakdown of Success Rates across 9 application categories and the composite Workflow category, offering empirical insight into the current capability boundaries of GUI agents. The data reveal that the current SOTA performance is a total SR of 69.9% achieved by Agent S3 (Gonzalez-Pumariega et al., 2025). Other advanced models, such as GTA1 and Claude Sonnet 4.5, achieve SRs over 60%. A critical insight from Table 6 is the stark performance disparity between single-application tasks and complex workflows. In isolated, domain-specific environments, agents demonstrate high competence. However, performance significantly degrades in the Workflow category, which involves multi-application tasks. The SOTA SR drops to 58.7% in this category, highlighting the difficulty of open-world tasks. ReAct, Reflexion, and their variants are widely adopted by top-performing agents. Agentic Lybic (Guo et al., 2025a) and CoAcT-1 (Song et al., 2025) both implement multi-agent systems, and Test-Time Scaling has been adopted by GTA1 and Agent S3. These advances improve decision quality in complex environments, enabling agents to perform competitively on benchmarks such as OSWorld.

### 3.4 Systematic Audit of Benchmark Coverage, Comparability, and Limitations

GUI agent evaluation spans benchmark families with qualitatively different task forms. Grounding settings are relatively close to CV-style perception tests, whereas navigation and open-world settings are closer to interactive evaluation, where observations, action spaces, execution conditions, and evaluators all shape what is being measured. Table 7 therefore uses a common audit schema to clarify benchmark families, comparison scope, and major coverage gaps.

**Coverage Audit.** Table 7 reveals that component-level capabilities have been benchmarked much more extensively. Grounding benchmarks dominate static settings, while navigation and context tracking become more central as benchmarks move toward long-horizon or executable evaluation. However, system-level coverage remains uneven. Adaptation and generalization appear relatively often, but safety, personalization, efficiency, and direct human-utility signals remain concentrated in a small number of specialized benchmarks. This fragmentation is itself informative: many GUI-agent benchmarks evaluate different capability bundles under different deployment assumptions rather than different difficulty levels of one shared task.

Table 7: Representative cross-benchmark audit of GUI-agent evaluation under a coarse coding scheme.

| Benchmark | Benchmark setup | | | Evaluator family | | | Component-level | | | System-level | | | |
|---|---|---|---|---|---|---|---|---|---|---|---|---|---|
| | Env. | Obs. | Act. | Traj. | Diag. | Hum. | Grd. | Nav. | Ctx. | Adp. | Saf./Priv. | Pers. | Eff. |
| ScreenSpot-V2 | Sta | V | Loc | – | – | – | √ | – | – | – | – | – | – |
| ScreenSpot-Pro | Sta | V | Loc | – | – | – | √ | – | – | – | – | – | – |
| GUI-World | Sta | Vid | QA | – | △ | △ | △ | – | △ | √ | – | – | – |
| TransBench | Sta | V | Loc | – | – | – | √ | – | – | √ | – | – | – |
| Mind2Web | Sta | S | Trace | √ | – | – | √ | √ | √ | √ | – | – | – |
| AITW | Sta | V+S | Trace | √ | – | – | √ | √ | △ | √ | – | – | – |
| WebArena | Sim | V+S | Exec | △ | – | △ | √ | √ | △ | – | – | △ | – |
| MT-Mind2Web | Sim | S | Trace | √ | – | – | √ | √ | √ | √ | – | – | – |
| GUI-Odyssey | Sim | V | Trace | √ | – | – | √ | √ | √ | √ | – | – | – |
| OSWorld | Real | V+S | Exec | – | – | △ | √ | √ | √ | △ | – | – | – |
| AndroidWorld | Real | V+S | Exec | – | – | △ | √ | √ | √ | √ | – | – | – |
| CRAB | Real | V | Exec | – | √ | △ | √ | √ | √ | √ | – | – | √ |
| WindowsAgentArena | Real | V+S | Exec | – | – | △ | √ | √ | √ | △ | △ | – | – |
| ST-WebAgentBench | Real | V+S | Exec | △ | √ | – | √ | √ | △ | △ | √ | △ | – |
| FingerTip | Real | V+S | Exec | √ | △ | √ | √ | √ | √ | △ | △ | √ | √ |

Symbols: √ direct/primary coverage; △ partial or indirect coverage; – not a main evaluation target.
Environment (Env.): Sta = non-interactive static or trace-based setting; Sim = executable simulated environment; Real = executable real OS/mobile/web environment.
Available observations (Obs.): V = screenshots; S = structural input such as DOM or A11y trees; Vid = video or temporally extended visual input.
Action executability (Act.): Loc = element localization; QA = question answering / information extraction; Trace = offline action-sequence prediction on recorded traces; Exec = executable interaction.
Evaluator family: the result-only column is omitted. Traj. = trajectory or action-sequence matching; Diag. = diagnostic evaluators such as termination analysis; Hum. = direct human-centered evaluation or explicit user-utility proxy signals.
Component-level targets: Grd. = GUI grounding; Nav. = multi-step navigation/action sequencing; Ctx. = context tracking across turns, steps, or user history. Basic intent understanding is omitted as a separate column.
System-level targets: Adp. = adaptation or generalization; Saf./Priv. = safety, trustworthiness, or privacy protection; Pers. = personalization or preference-sensitive execution; Eff. = computational or interaction efficiency.

**Comparability Audit.** The table also clarifies why cross-benchmark comparison remains fragmented.

- The information observable to the agent differs substantially. Some benchmarks are screenshot-only or video-only, some expose structural traces such as DOM or A11y trees, and some permit hybrid observation or richer environment APIs.

- The action spaces are mismatched, so the operational meaning of a successful prediction can range from coordinate localization, to step-level imitation, to end-to-end task completion in a live environment.

- The evaluator families differ and probe different failure modes. A benchmark that reports high final-task success may still hide poor step quality, unsafe behavior, redundant action loops, or excessive inference cost.

- Open-world benchmarks are closer to deployment conditions, yet they rely more heavily on environment setup, tool availability, policy definitions, or user-history assumptions. Hence, greater realism does not automatically imply stronger comparability.

These mismatches suggest a stricter reading rule for benchmark results. Scores should be interpreted primarily as within-family indicators. The most meaningful comparisons occur only when benchmarks share similar observability assumptions, action spaces, execution settings, evaluator families, and step or compute budgets. In this sense, comparability decreases as evaluation becomes more interactive: grounding benchmarks admit the strongest cross-model comparison, whereas open-world benchmarks are most informative as stress tests under specific deployment assumptions.

**Remaining Limitations.** Limitations of the current benchmark ecosystem remain. First, evaluation protocols are still not standardized enough for strong cross-paper conclusions. Real-OS benchmarks can be sensitive to environment setup, tool availability, step limits, compute budgets, and model versions. Released trajectories and widely circulated benchmark examples also create exposure or contamination risks that blur the boundary between genuine generalization and benchmark familiarity. In addition, LLM-as-a-judge evaluators are useful for open-ended tasks, but they introduce rubric sensitivity. Second, several deployment-relevant dimensions remain under-evaluated or are operationalized through heterogeneous proxies. Generalization is measured differently across benchmarks; explainability is rarely benchmarked directly; repeated interaction and long-term user modeling remain insufficiently covered; and benchmark-defined satisfaction or preference signals do not replace real human-centered studies. These limitations reinforce the need to read benchmark scores as family-specific evidence rather than as a single global ranking of GUI agents.

### 3.5 Observations

Taken together, the taxonomy, case studies, and systematic audit suggest four high-level observations about the current benchmark landscape. These observations are useful not only for interpreting existing results, but also for clarifying what future benchmark design should prioritize.

- *Benchmark Breadth Is Expanding Faster Than Evaluation Standardization.* The field has broadened rapidly and recent benchmarks increasingly attempt to cover system-level concerns such as safety, personalization, efficiency, and user-facing utility. Yet evaluation standardization has not kept pace with this expansion. Coverage of explainability, long-term interaction, and human-centered utility remains sparse. For benchmark designers, the implication is that expanding task breadth is not sufficient. Benchmark scope and reporting conventions must also become more systematic.

- *Realism Improves Ecological Validity but Requires Explicit Comparison Boundaries.* Moving from static to simulated to real environments makes evaluation more deployment-relevant, but it also changes what the agent can observe, which actions are available, and how success is judged. As a result, a benchmark score increasingly reflects a specific capability bundle under specific execution assumptions rather than a universal measure of GUI-agent quality.

- *Richer Automated Metrics Help, but Human-Centered Evaluation Remains a Missing Pillar.* Trajectory matching, termination analysis, efficiency, safety, and preference-aware signals provide a more informative picture than success rate alone. However, current benchmark practice still relies predominantly on automated scores, while deployment quality also depends on whether users understand the agent's behavior, can recover from failures, and actually benefit in practice.

- *The Main Deployment Gap Appears in Long-Horizon, Cross-Application Generalization.* Case studies show that performance drops sharply when tasks require longer horizons, cross-application coordination, or live execution under dynamic conditions. The clearest capability gap to practical deployment therefore lies not in closed-world competence. This suggests that the next generation of benchmarks should place greater weight on long-horizon, cross-application evaluation while making protocol assumptions and comparison boundaries explicit.

- Taken together, these observations suggest that the central challenge is no longer merely to build harder GUI benchmarks, but to design benchmark families whose coverage, comparison scope, and connection to real user utility are all more explicit.

## 4 Challenges and Future Directions

Despite the rapid evolution of GUI agents, significant discrepancies remain between benchmark performance and practical deployment. We synthesize these challenges into six key dimensions, highlighting the trade-offs in current methodologies and the necessary shifts for future research.

### 4.1 Lack of Explainability Evaluation

While GUI agent evaluation has expanded significantly across multiple dimensions, as reflected in Section 3, the systematic assessment of decision explainability remains notably underdeveloped. Users currently need insight into why an agent makes specific decisions, particularly during failures or sensitive operations. Evaluation frameworks should integrate explainability. This could involve requiring agents to provide natural language justifications for key decisions, designing tasks that test logical consistency across multi-step plans, or incorporating interactive mechanisms that allow users to query the reasoning post hoc. This is an important step toward building GUI agents that are transparent and trustworthy in sensitive environments.

### 4.2 The Challenge of Generalization

Despite the rapid evolution of GUI agents, significant gaps remain between benchmark performance and practical deployment. These gaps are closely related to the broader goal of data-efficient agentic learning: enabling agents to learn, adapt, and make decisions under limited supervision, feedback, and interaction budgets (Wang et al., 2026). We synthesize the remaining challenges into six key dimensions.

### 4.3 Security and Trustworthiness Vulnerabilities

As agents are granted extensive permissions to operate devices and access private data of users, security has become an important concern. Most existing benchmarks focus on functional completion and barely assess the ability to identify and reject harmful or illicit instructions. While pioneering benchmarks like ST-WebAgentBench and MobileSafetyBench have emerged, current agents exhibit weak defenses against real-world threats, such as indirect prompt injection, UI forgery, and malicious overlays. Trustworthiness must be integrated into the core evaluation, pushing agents to develop robust safety and privacy protection mechanisms.

### 4.4 Human-Centered Evaluation Beyond Automated Scores

Automated metrics remain indispensable, but they do not directly measure whether users trust the agent, understand its behavior, recover from failures, or actually benefit in practice. Even user-oriented signals such as Satisfaction Score or proactive and personalized tracks are still benchmark-defined proxies. Future work should therefore include human-in-the-loop studies of usability, practical efficiency, recovery behavior, and oversight quality (Chen et al., 2025a).

### 4.5 Personalized User Modeling

Current benchmarks predominantly evaluate standardized, isolated tasks, neglecting the personalized nature of human-machine interaction. They rarely assess whether GUI agents can learn user habits, maintain long-term memory, or adapt to personalized constraints over repeated interactions. A truly helpful assistant must evolve from passive command execution to proactive suggestions and personalized workflows. Building on pioneers such as FingerTip, future benchmarks can make this dimension more explicit by evaluating repeated interactions, user-profile adaptation, and preference-sensitive tasks.

### 4.6 Light-Weighting and On-Device Deployment

The successful adoption of GUI agents in consumer-facing applications necessitates a rigorous assessment of their on-device deployment feasibility. This transition introduces a core challenge: how to effectively miniaturize large LLMs/MLLMs (as SOTA agents often rely on models above 7B) to meet strict computational and latency constraints. Model compression often results in significant performance degradation, particularly in core functions such as precise grounding and planning. While existing benchmarks have started to include efficiency metrics like Time/Token Costs, more critical physical constraints for on-device deployment, such as memory/VRAM footprint and power consumption, remain largely undiagnosed.

## 5    Conclusion

GUI agents represent a pivotal step towards Artificial General Intelligence (AGI), bridging the gap between digital systems and human intentions. This survey has provided a comprehensive review of the rapid evolution of LLM-based GUI agent benchmarks, tracing the trajectory from grounding and QA tasks to multi-step navigation tasks, and finally to the complex, dynamic open-world environments that define the current frontier. Our analysis reveals a clear co-evolutionary relationship: advancements in LLMs and MLLMs empower agents with stronger capabilities, which in turn necessitate benchmarks with higher fidelity, broader coverage, and stricter evaluation standards. The future of this field relies not just on building stronger models, but also on establishing rigorous, reproducible, and realistic benchmarks that can accurately measure the gap between autonomous agents and human users.

For researchers who aim to design new benchmarks, the main takeaway is straightforward: first determine the benchmark family, then specify the dataset, environment, and evaluator, report capability coverage and multi-dimensional metrics, formalize generalization conditions, and make clear which comparisons are intended to be meaningful. We hope this survey serves as a roadmap for researchers and developers, illuminating the current landscape and guiding the community towards creating GUI agents that are not only intelligent but also efficient, trustworthy, and truly capable of assisting users in the complex digital world.

## Acknowledgment

This work is supported by Beijing Science and Technology Program (No.Z251100008125003).

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
