# OpenReview forum: "A Survey on Benchmarks of LLM-based GUI Agents"
_TMLR — Accepted by TMLR_

### Review · Reviewer_qMbi · 2026-03-26

**Summary Of Contributions:**

This submission surveys benchmarks for LLM-based GUI agents, organizing the space into three task-scenario categories: grounding/QA, navigation, and open-world evaluation. It also proposes a benchmark-design decomposition into dataset, environment, and evaluator, and discusses both component-level capabilities such as grounding, navigation, and context tracking, and system-level concerns such as adaptation, privacy, personalization, and efficiency. The paper’s main strengths are that it addresses a timely topic, offers a fairly clear taxonomy, and compiles a broad set of benchmark tables spanning web, mobile, desktop, and cross-platform settings. The paper also surfaces several useful future directions, especially around safety, generalization, personalization, and on-device deployment.

**Audience:**

Yes

**Audience Explanation:**

I expect this paper would be useful to researchers working on agents, multimodal systems, evaluation, and human-computer interaction. GUI agents are developing quickly, and benchmark fragmentation is a real issue. A survey that consolidates benchmark types, environments, and metrics is likely to be valuable even if the contribution is mainly organizational. This is also squarely within TMLR’s scope, which explicitly includes surveys that draw connections, highlight trends, and suggest new problems.

**Broader Impact Concerns:**

The paper already touches on important risk-related issues, especially privacy protection, malicious interfaces, prompt injection, and safety-aware evaluation, and I appreciated that these are treated as benchmark dimensions rather than afterthoughts.

**Claims And Evidence:**

Yes

**Claims Explanation:**

For the paper’s main descriptive claims, the evidence is mostly adequate. The taxonomy is consistently reflected in the organization of Sections 3.1–3.3, and the benchmark tables do support the claim that the field has evolved from static grounding/QA settings to navigation and then open-world environments. The paper also provides a coherent conceptual framing of benchmark design through dataset, environment, and evaluator, and this framing aligns reasonably well with the examples discussed throughout the survey.

My reservations are about rigor and calibration rather than correctness. The paper presents itself as a systematic review, but I did not see a sufficiently explicit methodology for literature collection and inclusion/exclusion. That makes it harder to assess completeness and harder to know whether omissions are principled or accidental. Relatedly, the “case study” sections often function more as narrative snapshots plus leaderboard excerpts than as genuine comparative analyses. Some claims about trends in architectural evolution or benchmark progress would be stronger if the paper more clearly separated apples-to-apples comparisons from headline numbers collected under different settings. Finally, a few novelty-positioning statements, especially the “first” benchmark-centric survey claim, should either be substantiated more carefully or softened.

**Requested Changes:**

1. The paper should describe a clearer survey methodology. Please state how papers/benchmarks were collected, the time window covered, and the inclusion and exclusion criteria. Without this, the survey feels informative but not fully systematic.

2. The synthesis should be strengthened beyond enumeration. In particular, the paper should more explicitly discuss where cross-benchmark comparisons are not directly meaningful because environments, action spaces, metrics, or evaluation protocols differ. This would make the survey more trustworthy and more useful.

3. The survey appears to omit some relevant recent computer-use-agent work, for example, OpenCUA, which seems closely related to the paper’s open-world evaluation discussion. Even if the authors view it as more of a foundation/model paper than a benchmark paper, it would be helpful to clarify the inclusion criteria and discuss borderline cases like this.

4. There are frequent grammar problems and typos, for instance: Page 6 "These metrics helps researcher"; Page 10 "was also a notable examples"; Page 7 "is a recent advancement of ScreenSpot which focus on"

---

> ### Author Response · Authors · 2026-04-21
> **Response to Reviewer qMbi (Part 1/2)**
>
> ## General response summary
>
> We thank the reviewer for the thoughtful and constructive review. We appreciate the reviewer’s recognition that the paper addresses a timely topic, provides a clear taxonomy, compiles broad benchmark tables, and highlights useful directions such as safety, generalization, personalization, and on-device deployment. We agree with the reviewer’s main concerns about survey methodology, synthesis beyond enumeration, cross-benchmark comparability, omission of relevant recent computer-use-agent work such as OpenCUA, novelty calibration, leaderboard interpretation, and grammar issues.
>
> In the revised manuscript, we made the following major changes:
>
> 1. We added a “Survey Scope and Methodology” paragraph in the Introduction, clarifying how papers and benchmarks were collected, the time scope, inclusion/exclusion criteria, representative benchmark selection, and the use of leaderboard snapshots.
> 2. We added a new Section 3.4, “Systematic Audit of Benchmark Coverage, Comparability, and Limitations,” together with Table 7, to provide synthesis beyond enumeration and to separate benchmark interpretation from uncontrolled cross-benchmark comparison.
> 3. We expanded the discussion of cross-benchmark comparability, including environment differences, action-space differences, evaluation-protocol differences, observability assumptions, and benchmark-family-specific interpretation.
> 4. We added OpenCUA-related discussion and included AgentNetBench in the navigation benchmark table. We also cite OpenCUA in the discussion of SFT and include OpenCUA-72B in the OSWorld leaderboard snapshot.
> 5. We softened the novelty positioning and now frame the paper as an evaluation-centric complement to existing method-centric surveys.
> 6. We carefully proofread the manuscript and corrected grammar and wording issues, including the examples pointed out by the reviewer.
>
> ## Point-wise replies
>
> ### Comment 1:
> “The paper should describe a clearer survey methodology. Please state how papers/benchmarks were collected, the time window covered, and the inclusion and exclusion criteria. Without this, the survey feels informative but not fully systematic.”
>
> ### Response:
> We agree and have added a new “Survey Scope and Methodology” paragraph in the Introduction. The revised paragraph states that we focus on publicly available benchmarks and evaluation resources for LLM-based GUI agents across web, mobile, desktop, and cross-platform settings. It explains that candidate papers and benchmark resources were collected from arXiv, GitHub repositories, relevant journals and conference venues, and backward/forward citation tracing. It also states that we mainly consider benchmark papers published since 2023 while including selected influential earlier works.
>
> We further clarify inclusion and exclusion criteria. A work is included as a core benchmark entry if it introduces a benchmark, dataset, environment, evaluator, or standardized evaluation setup directly used to assess GUI agents. Papers whose primary contribution is agent architecture or training without a distinct benchmark contribution are cited when they provide necessary context, but are not treated as core benchmark entries. We also explain how representative benchmarks are selected for summary tables and case studies. These revisions make the search sources, time scope, inclusion/exclusion criteria, selection rationale, and treatment of agent-method papers versus benchmark papers more explicit.
>
> ### Comment 2:
> “The synthesis should be strengthened beyond enumeration. In particular, the paper should more explicitly discuss where cross-benchmark comparisons are not directly meaningful because environments, action spaces, metrics, or evaluation protocols differ. This would make the survey more trustworthy and more useful.”
>
> ### Response:
> We agree and have strengthened the synthesis substantially. We added a new Section 3.4 and Table 7. Table 7 audits representative benchmarks under a common schema, including environment type, available observations, action executability, evaluator family, component-level targets, and system-level targets. This table is intended to make benchmark coverage and comparison boundaries explicit.
>
> The accompanying text now explains why cross-benchmark comparisons are often not directly meaningful. For example, benchmarks differ in whether agents observe screenshots, videos, DOM/A11y trees, or hybrid inputs; whether actions are localization predictions, offline trace predictions, or executable interactions; and whether evaluation uses final success, trajectory matching, diagnostic termination analysis, user-utility proxies, or LLM-as-a-judge. We also state that benchmark scores should primarily be interpreted within families that share similar observability assumptions, action spaces, execution settings, evaluator families, and step/compute budgets. Section 3.5 was also revised so that the observations are grounded in this audit.

---

> ### Author Response · Authors · 2026-04-21
> **Response to Reviewer qMbi (Part 2/2)**
>
> ### Comment 3:
> “The survey appears to omit some relevant recent computer-use-agent work, for example, OpenCUA, which seems closely related to the paper’s open-world evaluation discussion. Even if the authors view it as more of a foundation/model paper than a benchmark paper, it would be helpful to clarify the inclusion criteria and discuss borderline cases like this.”
>
> ### Response:
> We agree and have added OpenCUA-related content. In the revised manuscript, we include AgentNetBench from OpenCUA in Table 3 as a navigation benchmark entry. We also discuss OpenCUA in Section 3.2.2 as an example showing that scaling cross-OS demonstrations, compact state-action trajectories, and reflective long CoT supervision can strengthen computer-use agents. In addition, we include OpenCUA-72B in the OSWorld leaderboard in Table 6.
>
> We also clarified our inclusion criteria in the new “Survey Scope and Methodology” paragraph. The revised text explains that benchmark/dataset/environment/evaluator contributions are treated as core benchmark entries, while primarily architecture or training papers are cited when they provide necessary context for interpretation. This clarification is intended to make the treatment of borderline works more transparent, rather than making omissions appear accidental.
>
> ### Comment 4:
> “There are frequent grammar problems and typos, for instance: Page 6 ‘These metrics helps researcher’; Page 10 ‘was also a notable examples’; Page 7 ‘is a recent advancement of ScreenSpot which focus on’”
>
> ### Response:
> We agree and have carefully proofread the manuscript. We corrected the examples mentioned by the reviewer and many other grammar and wording issues throughout the paper. For instance, “These metrics helps researcher” was revised to “These metrics help researchers,” “was also a notable examples” was revised to “was another notable benchmark,” and “is a recent advancement of ScreenSpot which focus on” was revised to “is a recent advancement of ScreenSpot that focuses on.” We also improved wording consistency and readability across the manuscript.
>
> ### Additional comment:
> “My reservations are about rigor and calibration rather than correctness. The paper presents itself as a systematic review, but I did not see a sufficiently explicit methodology for literature collection and inclusion/exclusion. That makes it harder to assess completeness and harder to know whether omissions are principled or accidental.”
>
> ### Response:
> We appreciate this clarification and agree that our original version needed better rigor and calibration. The new “Survey Scope and Methodology” paragraph was added specifically to clarify literature collection, time scope, inclusion/exclusion criteria, and representative benchmark selection. We also clarified that leaderboard results are illustrative snapshots rather than controlled cross-benchmark comparisons. We believe these changes make the completeness and scope of the survey more transparent.
>
> ### Additional comment:
> “Relatedly, the ‘case study’ sections often function more as narrative snapshots plus leaderboard excerpts than as genuine comparative analyses. Some claims about trends in architectural evolution or benchmark progress would be stronger if the paper more clearly separated apples-to-apples comparisons from headline numbers collected under different settings.”
>
> ### Response:
> We agree. In the revised manuscript, we now explicitly state in the methodology paragraph that leaderboard results in case studies are used mainly as illustrative snapshots rather than controlled cross-benchmark comparisons. We also added Section 3.4 to explain why apples-to-apples comparison is difficult across benchmark families. The new comparability audit discusses differences in observability, action spaces, evaluator families, execution settings, step or compute budgets, and protocol assumptions, and recommends interpreting scores mainly within comparable benchmark families. Section 3.5 was also revised to reflect these comparison boundaries. These changes help separate illustrative headline numbers from stronger within-family evidence.
>
> ### Additional comment:
> “Finally, a few novelty-positioning statements, especially the ‘first’ benchmark-centric survey claim, should either be substantiated more carefully or softened.”
>
> ### Response:
> We agree and have softened the novelty positioning. The revised manuscript now frames the paper as an evaluation-centric complement to existing method-centric GUI-agent surveys. The “Difference with Existing Surveys” paragraph has been rewritten to explain our specific contribution more carefully: organizing the literature by benchmark task scenarios, analyzing benchmarks through dataset/environment/evaluator pillars, and emphasizing persistent gaps between benchmark scores and real-world deployment. This removes the overstrong “first” framing and replaces it with a more precise explanation of how this survey differs from prior GUI-agent surveys.

---

### Review · Reviewer_wM1G · 2026-04-03

**Summary Of Contributions:**

The main contribution of this work is a review of the 'benchmarks' for LLM/VLM-based GUI agents. Most of the previous works have focused on reviewing the agent architectures directly while this work focuses more on critiquing the benchmarks that help evaluate agents for the GUI agents.

The paper is well-organized with a clear structure (pipeline, then evaluation, and then taxonomy). A central theme of the paper is that benchmarks are evolving alongside increasing task complexity, from grounding to navigation to open-world settings.

The main conclusion to note is that the current benchmarks still struggle to fully capture real-world performance and enable consistent comparison across methods.

**Audience:**

Yes

**Audience Explanation:**

The paper addresses the emerging area of LLM-based GUI agents and, more importantly, their evaluation, which is of interest to a large audience.

**Broader Impact Concerns:**

Including a broader impacts discussion would strengthen the paper, given that GUI agents operate on user systems and raise important concerns around safety, privacy, and misuse.

**Claims And Evidence:**

Yes

**Claims Explanation:**

The progression from Grounding/QA to Navigation to Open-World tasks is conceptually clean and easy for readers to understand. The diagram on page 7 (Figure 3) makes this progression visually clear by aligning it with increasing flexibility and difficulty, and that helps the paper tell a coherent story about the maturation of the field.

**Requested Changes:**

Benchmarks for GUI agents evaluate performance using a combination of result-based metrics (e.g., success rate), trajectory-level metrics (e.g., action matching), perception metrics (e.g., IoU), and more recent measures such as efficiency, termination analysis, and safety. However, these evaluations are often conducted in simplified or simulated environments and rely on heterogeneous metrics, making it difficult to consistently compare methods or reflect real-world usability. As a result, strong benchmark performance does not always translate to robust behavior in dynamic, cross-application, or real OS settings.

Despite identifying the gaps and highlighting them the paper falls short in performing a deeper analysis on the lack of standardized evaluation protocols, limited assessment of generalization, and missing dimensions like explainability, personalization, and long-term interaction.

Moreover, while the paper discusses many evaluation dimensions (success rate, trajectories, efficiency, safety), it focuses almost entirely on automated metrics and benchmark-driven evaluation. It does not meaningfully address evaluation with real users, such as:
- Whether users can trust or understand the agent’s behavior
- How usable or frustrating the interaction is
- Whether agents actually improve task efficiency in practice
- How users respond to errors, ambiguity, or recovery behavior

My point is that this paper never frames human evaluation as a missing pillar alongside dataset, environment, and evaluator. It does not discuss user studies, UX metrics, or human-in-the-loop evaluation. There is also no critique of automated metrics as proxies for human utility.

A limitation of this work is that, while it highlights fragmentation across benchmarks, it does not provide a *systematic audit* or detailed analysis of the factors needed to standardize evaluation protocols.

---

> ### Author Response · Authors · 2026-04-21
> **Response to Reviewer wM1G (Part 1/3)**
>
> ## General response summary
>
> We thank the reviewer for the positive and constructive review. We appreciate the reviewer’s recognition that the paper is well organized, that the grounding-to-navigation-to-open-world progression is conceptually clear, and that the paper addresses an emerging topic of interest to the TMLR audience. We agree with the reviewer’s main concerns that the original version needed deeper analysis of standardization, generalization, explainability, personalization, long-term interaction, human-centered evaluation, and broader safety/privacy implications.
>
> In the revised manuscript, we made the following major changes:
>
> 1. We added a new Section 3.4, “Systematic Audit of Benchmark Coverage, Comparability, and Limitations,” together with Table 7, to provide a more systematic analysis of benchmark coverage and standardization challenges.
> 2. We expanded the discussion of benchmark comparability, including differences in observability, action spaces, evaluator families, execution settings, step/compute budgets, real-OS reproducibility, and the fidelity-reproducibility trade-off.
> 3. We added human-centered evaluation as an explicit emerging direction in the Abstract and added a dedicated Section 4.4, “Human-Centered Evaluation Beyond Automated Scores.”
> 4. We revised Section 3.5 to explicitly state that human-centered evaluation remains a missing pillar, because current benchmark-defined satisfaction or preference signals are only proxies for real user utility.
> 5. We separated human-centered evaluation from personalization, adding Section 4.4 on human-centered evaluation beyond automated scores and strengthening Section 4.5 on personalization, long-term user modeling, and repeated interaction.
> 6. We strengthened the safety, privacy, and trustworthiness discussion in Section 4.3 and related benchmark discussions, including malicious interfaces, prompt injection, privacy risks, and the need for human oversight.

---

> ### Author Response · Authors · 2026-04-21
> **Response to Reviewer wM1G (Part 2/3)**
>
> ## Point-wise replies
>
> ### Comment 1:
> “Benchmarks for GUI agents evaluate performance using a combination of result-based metrics (e.g., success rate), trajectory-level metrics (e.g., action matching), perception metrics (e.g., IoU), and more recent measures such as efficiency, termination analysis, and safety. However, these evaluations are often conducted in simplified or simulated environments and rely on heterogeneous metrics, making it difficult to consistently compare methods or reflect real-world usability. As a result, strong benchmark performance does not always translate to robust behavior in dynamic, cross-application, or real OS settings.”
>
> ### Response:
> We agree with this important observation. In the revised manuscript, we added a new systematic audit section to address exactly this issue. Section 3.4 now explains that GUI-agent benchmarks differ not only in difficulty but also in benchmark family, environment type, observability assumptions, action spaces, and evaluator families. We explicitly state that scores should be interpreted primarily within comparable benchmark families, because static grounding benchmarks, simulated navigation benchmarks, and real-OS open-world benchmarks measure different capability bundles.
>
> We also added a stronger discussion of the gap between benchmark performance and real-world deployment. In particular, Section 3.4 explains why moving toward more realistic settings makes evaluation more deployment-relevant but reduces straightforward comparability, and Section 3.5 highlights long-horizon, cross-application generalization as a key deployment gap. These changes add a systematic analysis of heterogeneous metrics, execution settings, comparability boundaries, and why benchmark success may not imply robust real-world usability.
>
> ### Comment 2:
> “Despite identifying the gaps and highlighting them the paper falls short in performing a deeper analysis on the lack of standardized evaluation protocols, limited assessment of generalization, and missing dimensions like explainability, personalization, and long-term interaction.”
>
> ### Response:
> We agree that the original version identified these issues but did not analyze them systematically enough. The revision addresses this in two ways.
>
> First, Section 3.4 and Table 7 provide a cross-benchmark audit showing which benchmarks cover component-level capabilities and which cover system-level dimensions. This makes it clearer that system-level dimensions such as safety/privacy, personalization, efficiency, and human-centered utility remain unevenly covered. The audit also shows why evaluation standardization is difficult: benchmarks differ in environment, observation channels, action executability, evaluator families, and deployment assumptions.
>
> Second, we revised the future-directions section to separate several missing dimensions more clearly. Section 4.1 discusses explainability evaluation; Section 4.2 discusses generalization; Section 4.4 discusses human-centered evaluation beyond automated scores; Section 4.5 discusses personalized user modeling and repeated interaction; and Section 4.6 discusses lightweight and on-device deployment. In particular, we separated human-centered evaluation from personalization so that user studies, usability, practical task efficiency, recovery behavior, and oversight quality are treated as distinct evaluation concerns rather than folded into personalization. These revisions make explainability, generalization, human-centered evaluation, personalization, long-term interaction, and deployment constraints more explicit.

---

> ### Author Response · Authors · 2026-04-21
> **Response to Reviewer wM1G (Part 3/3)**
>
> ### Comment 3:
> “Moreover, while the paper discusses many evaluation dimensions (success rate, trajectories, efficiency, safety), it focuses almost entirely on automated metrics and benchmark-driven evaluation. It does not meaningfully address evaluation with real users, such as:
> Whether users can trust or understand the agent’s behavior
> How usable or frustrating the interaction is
> Whether agents actually improve task efficiency in practice
> How users respond to errors, ambiguity, or recovery behavior”
>
> ### Response:
> We agree and have revised the paper to explicitly address human-centered evaluation. The Abstract now lists human-centered evaluation as an emerging direction. In Section 3.3.1, we added a discussion noting that user-oriented benchmark signals, such as satisfaction scores or proactive/personalized tracks, are still only partial proxies for human utility. In Section 3.5, we added an observation titled “Richer Automated Metrics Help, but Human-Centered Evaluation Remains a Missing Pillar,” emphasizing that deployment quality depends on whether users understand the agent’s behavior, can recover from failures, and actually benefit in practice.
>
> We also added a dedicated Section 4.4, “Human-Centered Evaluation Beyond Automated Scores.” This section states that automated metrics and benchmark-defined proxies, such as satisfaction scores or proactive/personalized tracks, do not directly measure user trust, understandability, failure recovery, practical benefit, or oversight quality. It recommends human-in-the-loop studies of usability, practical efficiency, recovery behavior, and oversight quality.
>
> ### Comment 4:
> “My point is that this paper never frames human evaluation as a missing pillar alongside dataset, environment, and evaluator. It does not discuss user studies, UX metrics, or human-in-the-loop evaluation. There is also no critique of automated metrics as proxies for human utility.”
>
> ### Response:
> We agree with this critique and have directly addressed it. The revised manuscript now explicitly frames human-centered evaluation as a missing pillar. Table 7 includes a “Hum.” column under evaluator family to indicate whether a benchmark contains direct human-centered evaluation or explicit user-utility proxy signals. Section 3.5 states that current benchmark practice still relies predominantly on automated scores, while real deployment quality also depends on user understanding, recoverability, and practical benefit. Section 4.4 further critiques automated benchmark scores as incomplete proxies for human utility.
>
> ### Comment 5:
> “A limitation of this work is that, while it highlights fragmentation across benchmarks, it does not provide a systematic audit or detailed analysis of the factors needed to standardize evaluation protocols.”
>
> ### Response:
> We agree and added a systematic audit. The new Table 7 compares representative benchmarks using a coarse but explicit coding scheme. It covers environment type, observations, action executability, evaluator family, component-level targets, and system-level targets. The accompanying Section 3.4 discusses coverage gaps, comparability barriers, and remaining limitations of current benchmark protocols. We also added a practical takeaway in the conclusion: future benchmark designers should specify the benchmark family, dataset, environment, evaluator, capability coverage, multidimensional metrics, generalization conditions, and intended comparison scope.
>
> ### Comment 6:
> “Including a broader impacts discussion would strengthen the paper, given that GUI agents operate on user systems and raise important concerns around safety, privacy, and misuse.”
>
> ### Response:
> We agree that safety, privacy, and misuse are especially important for GUI agents because they operate directly on user systems and may access sensitive information. In the revised manuscript, we strengthened these issues throughout the paper. Section 2.2 discusses privacy protection as a system-level capability, including rejection of harmful instructions and resistance to malicious threats such as UI forgery, malicious pop-ups, overlays, and indirect prompt injection. Section 3.3.1 discusses safety-oriented benchmarks such as ST-WebAgentBench, MobileSafetyBench, AgentScan, and macOSWorld. Section 4.3 further emphasizes security and trustworthiness vulnerabilities, including access to private data, harmful or illicit instructions, and malicious interfaces such as prompt injection or UI forgery, and stresses the need to integrate safety and privacy protection into core evaluation. Section 4.4 also discusses human oversight quality as part of human-centered evaluation.

---

### Review · Reviewer_9JkC · 2026-04-08

**Summary Of Contributions:**

This paper presents a survey of benchmarks for LLM-based GUI agents and argues that evaluation has evolved from single-step Grounding & QA tasks to Navigation tasks and then to Open-World environments. Its main contributions are:

**(i)** a task-scenario taxonomy for benchmarking,

**(ii)** a decomposition of evaluation objectives into component-level capabilities such as intent understanding, grounding, navigation, and context tracking, plus system-level capabilities such as adaptation, personalization, privacy protection, and computational efficiency,

**(iii)** a curated overview of benchmark datasets/environments/evaluators across grounding, navigation, and open-world settings, and

**(iv)** a discussion of open challenges including explainability, generalization, trustworthiness, personalization, and lightweight deployment. The paper also uses representative case studies and leaderboard snapshots such as ScreenSpot-Pro, GUI-Odyssey, and OSWorld to illustrate progress and remaining gaps.

**Additional Comments:**

Overall, I found the paper timely, useful, and likely valuable to the community. The taxonomy is easy to follow, the benchmark tables are helpful, and the emphasis on open-world evaluation, safety, and generalization is appropriate. My main reservation is not about relevance, but about rigor of positioning: the manuscript should be more careful about its novelty claim and more explicit about survey methodology. With those changes, I think it would become a substantially stronger survey.

**Audience:**

Yes

**Audience Explanation:**

This is a timely topic for the TMLR audience because GUI agents sit at the intersection of LLMs/MLLMs, agents, benchmarking, and human-computer interaction, and the benchmark landscape is moving very quickly. Researchers working on multimodal agents, evaluation, interactive systems, and deployment would likely find this survey useful as a structured overview of the space, especially because it organizes benchmarks by task scenario and highlights where current systems still fail in realistic settings. Even readers who already know GUI agents broadly may benefit from the benchmark tables and the paper’s synthesis of open-world evaluation, safety, and generalization issues.

**Broader Impact Concerns:**

I do not have major broader-impact concerns specific to the survey genre.

**Claims And Evidence:**

Yes

**Claims Explanation:**

The survey’s descriptive claims are generally supported well. The paper clearly defines benchmark components and evaluation objectives, then backs its taxonomy with concrete benchmark tables for Grounding & QA, Navigation, and Open-World settings. It also provides case studies with concrete numbers, such as ScreenSpot-Pro for grounding, GUI-Odyssey for long-horizon navigation, and OSWorld for open-world execution, which makes the discussion readable and evidence-based. The observations section also reasonably synthesizes patterns such as the move from static to dynamic environments and the increasing use of richer metrics beyond plain success rate.

That said, I think the paper’s strongest positioning claim is less convincingly supported. The manuscript states that it is the first systematic survey from the perspective of benchmarks and evaluation methodologies, but prior GUI-agent surveys already explicitly cover benchmarks and evaluation metrics, even if they are not benchmark-centered to the same degree. For example, GUI Agents: A Survey says it categorizes “benchmarks, evaluation metrics, architectures, and training methods,” and Large Language Model-Brained GUI Agents: A Survey has a full evaluation section covering metrics, measurements, evaluation platforms, and platform-specific benchmarks. So I would encourage the authors to soften the novelty claim and position this paper more precisely as a benchmark-centric or evaluation-centered survey rather than the first systematic review of benchmarks outright.

I also think some of the higher-level conclusions are still more narrative than systematic. Claims about “co-evolution,” “shift toward multi-faceted evaluation,” and the “persistent generalization gap” are plausible and mostly correct, but they are not derived from an explicit survey protocol, inclusion/exclusion criteria, cutoff date, or quantitative meta-analysis of the surveyed benchmarks. In addition, some leaderboard comparisons are aggregated from prior papers rather than unified re-evaluation, which is informative but not fully controlled.

**Requested Changes:**

**I** the authors should clarify the paper’s novelty relative to existing surveys. The current “first benchmark-centric survey” claim feels overstated given prior surveys that already cover benchmarks and evaluation. I would recommend rewriting this comparison section to explain exactly what is new here: for example, deeper benchmark organization, stronger emphasis on evaluators/metrics, or a more benchmark-centric taxonomy.

**II** the paper should include a more explicit survey methodology: search scope, time cutoff, inclusion/exclusion criteria, and how representative benchmarks were selected. For a survey paper, this would make the review process more reproducible and would strengthen confidence in completeness.

**III** the paper should make a cleaner distinction between benchmark analysis and agent-technique review. In several case-study sections, the discussion shifts substantially toward SFT, RL, ReAct, MAS, and test-time scaling. Those are relevant, but they sometimes dilute the benchmark-centered framing. I would prefer either a more explicit rationale for why these technique discussions are necessary for benchmark interpretation, or a tighter refocus on what each benchmark uniquely measures.

**IV** the paper would be stronger with more systematic synthesis, not just narrative synthesis. Examples include: a matrix mapping each benchmark to capabilities and failure modes; counts over time by environment type (static/simulated/real); a clearer taxonomy of evaluators; or a summary of which benchmarks assess safety, personalization, efficiency, and zero-shot transfer. The observations section already points in this direction, but a more explicit comparative framework would make the paper more rigorous.

**V** I would like a stronger discussion of benchmark comparability and limitations, such as differences in action spaces, observability assumptions (visual-only vs DOM/A11y access), reproducibility in real OS settings, possible contamination from public tutorials/trajectories, and the reliability of judge-based evaluation.

**VI** the paper would benefit from a careful proofreading pass. There are multiple grammatical/wording issues throughout the draft, and polishing the prose would improve readability and authority.

---

> ### Author Response · Authors · 2026-04-21
> **Response to Reviewer 9JkC (Part 1/3)**
>
> ## General response summary
>
> We thank the reviewer for the positive and constructive assessment. We appreciate the reviewer’s recognition that the topic is timely, the taxonomy is useful, and the benchmark tables and case studies make the survey readable and evidence-based. We agree with the main concerns about novelty calibration, survey methodology, benchmark-centered framing, systematic synthesis, benchmark comparability, and writing quality.
>
> In the revised manuscript, we made the following major changes:
>
> 1. We softened the novelty claim and now position the paper as an evaluation-centric complement to existing method-centric GUI-agent surveys, rather than claiming to be the first systematic review outright.
> 2. We added a new “Survey Scope and Methodology” paragraph in the Introduction, clarifying search sources, citation tracing, the approximate time scope, inclusion/exclusion criteria, representative benchmark selection, and the role of leaderboard snapshots.
> 3. We revised the case-study framing so that discussions of SFT, RL, ReAct, MAS, and test-time scaling are explicitly motivated as context for interpreting benchmark results and limitations, rather than as a standalone agent-technique survey.
> 4. We added a new Section 3.4, “Systematic Audit of Benchmark Coverage, Comparability, and Limitations,” together with Table 7. This table provides a cross-benchmark audit over benchmark setup, evaluator family, component-level coverage, and system-level coverage, and makes explicit that GUI-agent benchmarks do not form a single homogeneous benchmark ladder.
> 5. We expanded the discussion of comparability and limitations, including observability assumptions, action-space mismatches, evaluator differences, real-OS reproducibility, contamination risks, LLM-as-a-judge reliability, and the need to read benchmark scores within benchmark families.
> 6. We carefully proofread the paper and corrected grammar, wording, and consistency issues throughout the manuscript.
>
> ## Point-wise replies
>
> ### Comment I:
> “The authors should clarify the paper’s novelty relative to existing surveys. The current ‘first benchmark-centric survey’ claim feels overstated given prior surveys that already cover benchmarks and evaluation. I would recommend rewriting this comparison section to explain exactly what is new here: for example, deeper benchmark organization, stronger emphasis on evaluators/metrics, or a more benchmark-centric taxonomy.”
>
> ### Response:
> We agree with this comment and have revised the positioning accordingly. In the Introduction, we removed the stronger “first systematic survey” framing and now describe our work as an “evaluation-centric complement” to existing method-centric GUI-agent surveys. We also rewrote the “Difference with Existing Surveys” paragraph to make the distinction more precise. The revised text explains that our contribution is not simply that we mention benchmarks, but that we organize the survey around task scenarios, analyze benchmarks through the three structural pillars of dataset, environment, and evaluator, and emphasize the gap between benchmark scores and real-world deployment. This replaces the earlier overstrong novelty claim with a more calibrated comparison to existing surveys.
>
> ### Comment II:
> “The paper should include a more explicit survey methodology: search scope, time cutoff, inclusion/exclusion criteria, and how representative benchmarks were selected. For a survey paper, this would make the review process more reproducible and would strengthen confidence in completeness.”
>
> ### Response:
> We agree that the original manuscript did not make the survey process explicit enough. We added a new “Survey Scope and Methodology” paragraph in the Introduction. It states that we focus on publicly available benchmarks and evaluation resources for LLM-based GUI agents across web, mobile, desktop, and cross-platform settings. It also explains that candidate papers and benchmark resources were collected from arXiv, GitHub, relevant journals and conferences, and backward/forward citation tracing. We further clarify that the survey mainly considers benchmark papers published since 2023 while including selected influential earlier works.
>
> We also define what counts as a core benchmark entry: a work is included if it introduces a benchmark, dataset, environment, evaluator, or standardized evaluation setup directly used to assess GUI agents. By contrast, papers whose primary contribution is agent architecture or training are cited for context but are not treated as core benchmark entries unless they introduce a distinct evaluation resource. Finally, we clarify that leaderboard results are used as illustrative snapshots rather than controlled cross-benchmark comparisons. These additions make the search scope, approximate time scope, inclusion/exclusion criteria, selection rationale, and limitations of leaderboard snapshots more explicit.

---

> ### Author Response · Authors · 2026-04-21
> **Response to Reviewer 9JkC (Part 2/3)**
>
> ### Comment III:
> “The paper should make a cleaner distinction between benchmark analysis and agent-technique review. In several case-study sections, the discussion shifts substantially toward SFT, RL, ReAct, MAS, and test-time scaling. Those are relevant, but they sometimes dilute the benchmark-centered framing. I would prefer either a more explicit rationale for why these technique discussions are necessary for benchmark interpretation, or a tighter refocus on what each benchmark uniquely measures.”
>
> ### Response:
> We agree that the previous case-study sections could be read as drifting toward a general agent-technique review. In the revision, we clarified why technique discussions are included: they help readers interpret what benchmark results reveal about agent capabilities and bottlenecks. For example, in Section 3.2.2, we now introduce SFT and RL specifically as training paradigms that help explain performance on long-horizon navigation benchmarks such as GUI-Odyssey. In Section 3.3.2, we revised the opening to explain that realistic open-world benchmarks expose capability bottlenecks and stimulate agent paradigms such as ReAct, Reflexion, MAS, and test-time scaling.
>
> We also tightened the wording in these sections to keep the emphasis on benchmark interpretation. The goal is not to survey all GUI-agent methods, but to explain why certain benchmark families require particular agent capabilities and why those benchmarks reveal remaining limitations.
>
> ### Comment IV:
> “The paper would be stronger with more systematic synthesis, not just narrative synthesis. Examples include: a matrix mapping each benchmark to capabilities and failure modes; counts over time by environment type (static/simulated/real); a clearer taxonomy of evaluators; or a summary of which benchmarks assess safety, personalization, efficiency, and zero-shot transfer. The observations section already points in this direction, but a more explicit comparative framework would make the paper more rigorous.”
>
> ### Response:
> We agree and have substantially strengthened the synthesis. We added a new Section 3.4, “Systematic Audit of Benchmark Coverage, Comparability, and Limitations,” and a new Table 7. Table 7 provides a cross-benchmark audit under a common coding scheme. It maps representative benchmarks to benchmark setup factors, including environment type, available observations, and action executability; evaluator family, including trajectory matching, diagnostic evaluation, and human-centered or user-utility proxy signals; component-level targets, including grounding, navigation, and context tracking; and system-level targets, including adaptation, safety/privacy, personalization, and efficiency.
>
> We chose this audit table because it directly addresses the reviewer’s request for a matrix-style synthesis and makes the coverage gaps easier to see. The accompanying text further discusses coverage patterns, comparability limitations, and remaining benchmark ecosystem limitations. It also clarifies the higher-level motivation for the audit: grounding benchmarks are relatively closer to CV-style comparison, whereas navigation and open-world benchmarks are closer to interactive evaluation, so they should not be treated as one homogeneous ranking ladder. We also revised Section 3.5 so that the observations are grounded in the audit rather than only narrative synthesis, and we expanded the observations into clearer benchmark-design implications.

---

> ### Author Response · Authors · 2026-04-21
> **Response to Reviewer 9JkC (Part 3/3)**
>
> ### Comment V:
> “I would like a stronger discussion of benchmark comparability and limitations, such as differences in action spaces, observability assumptions (visual-only vs DOM/A11y access), reproducibility in real OS settings, possible contamination from public tutorials/trajectories, and the reliability of judge-based evaluation.”
>
> ### Response:
> We agree, and this is now addressed explicitly in the revised Section 3.4. The new comparability audit discusses how benchmarks differ in the information observable to the agent, including screenshot-only, video-only, structural DOM/A11y traces, and hybrid observations. It also discusses mismatched action spaces, where success can mean coordinate localization, step-level imitation, or end-to-end executable task completion. We further clarify that evaluator families differ and can hide different failure modes, such as poor step quality, unsafe behavior, redundant action loops, or excessive inference cost.
>
> In the remaining-limitations paragraph, we also added discussion of how real-OS evaluation can be sensitive to environment setup and protocol conditions, including tool availability, step limits, compute budgets, and model versions. We added contamination risks from released trajectories and circulated benchmark examples, and we note that LLM-as-a-judge evaluation introduces rubric sensitivity. We now explicitly recommend that benchmark scores be read primarily as within-family indicators, rather than as one global ranking across all GUI agents.
>
> ### Comment VI:
> “The paper would benefit from a careful proofreading pass. There are multiple grammatical/wording issues throughout the draft, and polishing the prose would improve readability and authority.”
>
> ### Response:
> We agree and have carefully proofread the manuscript. We corrected grammar, subject-verb agreement, wording, capitalization, and consistency issues throughout the paper, including phrasing in the Introduction, benchmark descriptions, metric definitions, and case-study sections. We also revised several awkward sentences to improve readability and authority.
>
> ### Additional comment:
> “Overall, I found the paper timely, useful, and likely valuable to the community. The taxonomy is easy to follow, the benchmark tables are helpful, and the emphasis on open-world evaluation, safety, and generalization is appropriate. My main reservation is not about relevance, but about rigor of positioning: the manuscript should be more careful about its novelty claim and more explicit about survey methodology. With those changes, I think it would become a substantially stronger survey.”
>
> ### Response:
> We thank the reviewer for this encouraging assessment. The revision directly focuses on the two main reservations highlighted here. We calibrated the novelty claim and added a survey scope and methodology paragraph. We also went further by adding a systematic audit table and a detailed discussion of benchmark comparability and limitations. We believe these changes make the revised paper more careful in positioning and more rigorous as a benchmark-centered survey.

---

### Decision · Action_Editor_2Ys7 · 2026-05-20

**Recommendation:** Accept with minor revision

**Additional Comments:**

The review cycle spanned approximately several months, during which the GUI agent benchmark landscape continued to evolve rapidly. The authors may consider to conduct a targeted update of related work to incorporate relevant benchmark papers and evaluation resources that appeared or were made public during the submission-to-acceptance period.

**Audience:**

Yes

**Audience Explanation:**

The researchers on GUI agents would be interested in this paper.

**Claims And Evidence:**

Yes

**Claims Explanation:**

The topic is timely and the taxonomy (grounding/QA → navigation → open-world) is clear and well-supported
The authors satisfactorily addressed all major reviewer concerns in revision, including softening the novelty claim, adding a survey methodology paragraph, introducing a systematic benchmark audit (Section 3.4, Table 7), strengthening cross-benchmark comparability discussion, and adding a human-centered evaluation section

---

> ### Author Response · Authors · 2026-06-11
>
> Dear Action Editor,
>
> We have uploaded the camera-ready version of our paper.
>
> Following the suggestion, we conducted a targeted update of the related work and benchmark discussion to incorporate relevant GUI-agent benchmark papers and evaluation resources that appeared or became public during the submission-to-acceptance period. In particular, we updated the benchmark coverage and discussion to better reflect recent progress in GUI-agent evaluation, including newly released or recently public benchmark resources for memory-based GUI agents, dynamic GUI understanding, and mobile task evaluation.
>
> We also checked the final manuscript for camera-ready requirements and consistency.
>
> Thank you again for handling our submission and for the helpful suggestion on updating the rapidly evolving benchmark landscape.
>
> Best regards,
>
> The authors